_Article_

# Epiblast-derived CX3CR1+ progenitors generate cardiovascular cells during cardiogenesis

Kyuwon Cho[1], Mark Andrade [ID][1], Saeed Khodayari[1], Christine Lee [ID][1], Seongho Bae [ID][1], Sangsung Kim[2], Jin Eyun Kim[1] & Young-Sup Yoon [ID][1,2 ✉]

## Abstract

CX3CR1+ cells generate tissue macrophages in the developing heart and play cardioprotective roles in response to ischemic injuries in the adult heart. However, the origin and fate of CX3CR1+ cells during cardiogenesis remain unclear. Here, we performed genetic lineage tracing of CX3CR1+ cells and their progeny (termed Cx3cr1 lineage cells) in the mouse and demonstrated that they emerge from a subset of epiblast cells at embryonic day E6.5 and contribute to the parietal endoderm cells at E7.0. At E8.0–9.5 of development, Cx3cr1 lineage cells produced cardiomyocytes and endothelial cells via both de novo differentiation and fusion with pre-existing cardiomyocytes or endothelial cells, respectively. Cx3cr1 lineage cells persisted in the adult heart, comprising ~13% of cardiomyocytes and ~31% of endothelial cells. Additionally, CX3CR1+ cells differentiated from mouse embryonic stem cells generated cardiomyocytes, endothelial cells, and macrophages in vitro, ex vivo, and in vivo. Single-cell RNA sequencing revealed that Cx3cr1+ cells represent an intermediate cell population transitioning from embryonic stem cells to mesoderm. Taken together, embryonic CX3CR1+ cells constitute a multipotent epiblast-derived progenitor population that contributes not only to the formation of macrophages, but also of cardiomyocytes and endothelial cells.

**Key words** Cardiogenesis; CX3CR1; Cardiomyocyte; Endothelial Cells; Genetic Lineage Tracing

**Subject Categories** Development; Stem Cells & Regenerative Medicine

## Introduction

Ischemic heart disease is the primary cause of mortality in industrialized nations. Due to the limited regenerative capacity of the adult heart, no effective therapy is available for myocardial injury. Attempts to regenerate the injured heart via cell therapy resulted in limited and inconsistent benefits (Cahill, Choudhury et al, 2017). Thus, it becomes important to gain a better understanding of cardiac stem or progenitor cells (Li, He et al,

2018; Vagnozzi, Sargent et al, 2018; van Berlo et al, 2014) and their ontogenetic properties in order to develop an effective strategy for cardiac regeneration (van Berlo et al, 2014).

CX3CR1 (C-X3-C motif chemokine receptor 1) is a chemokine receptor for Fractalkine/CX3CL1 (C-X3-C motif chemokine ligand 1), which is involved in multiple biological and pathophysiological processes. It was first identified as an important mediator of migration and adhesion of immune cells, enabling their infiltration into the tissue (Imai, Hieshima et al, 1997; Lee, Lee et al, 2018). Its expression was also found in microglia (Nishiyori, Minami et al, 1998), neurons (Meucci, Fatatis et al, 2000), islet β cells (Lee, Morinaga et al, 2013), and smooth muscle cells (Lucas, Bursill et al, 2003). The function of CX3CR1 in tissue repair is controversial. Donnelly et al reported that genetic deletion of _Cx3cr1_ promoted recovery from spinal cord injuries (Donnelly, Longbrake et al, 2011). Ishida et al illustrated that genetic deletion of _Cx3cr1_ or treatment with anti-CX3CR1-neutralizing antibodies delayed skin wound healing (Ishida, Gao et al, 2008). For skeletal muscle regeneration, Arnold et al, showed that skeletal muscle repair was enhanced by genetic deletion of _Cx3cr1_ (Arnold et al, 2015) whereas Zhao et al, claimed the opposite (Zhao et al, 2016).

CX3CR1[+] cells play vital roles in cardiac development and disease. Developmentally, CX3CR1[+] cells originate from erythro-myeloid progenitors (EMPs) in the yolk sac (YS) at embryonic day 8.5 (E8.5) (Mass, Ballesteros et al, 2016; Schulz et al, 2012; Stremmel, Schuchert et al, 2018). They subsequently migrate to embryonic tissues, including the heart, and differentiate into tissue macrophages (Epelman, Lavine et al, 2014). The generation of tissue macrophages by CX3CR1[+] cells is independent of definitive hematopoiesis (Epelman et al, 2014; Schulz et al, 2012). Thus, CX3CR1[+] cells are regarded as macrophage progenitors (Epelman et al, 2014; Mass et al, 2016; Schulz et al, 2012; Yona, Kim et al, 2013). Embryonic cardiac macrophages are involved in efferocytosis, coronary vessel development, valve development, and lymphangiogenesis in the developing heart (Gula and Ratajska, 2022), making them an integral cellular component of cardiogenesis. These embryonic cardiac macrophages persist into the adult heart (Epelman et al, 2014) and play a cardioprotective role in the post-myocardial infarction (MI) heart by preventing adverse cardiac remodeling (Dick, Macklin et al, 2019). Despite the critical roles of CX3CR1[+] cells in heart development and disease, the origin and fate of CX3CR1[+] cells are unclear. Specifically, the existence of

---

[1]Department of Medicine, Division of Cardiology, Emory University School of Medicine, Atlanta, GA 30322, USA. [2]Severance Biomedical Science Institute, Yonsei University College of Medicine, Seoul, Republic of Korea. ✉E-mail: yyoon5@emory.edu

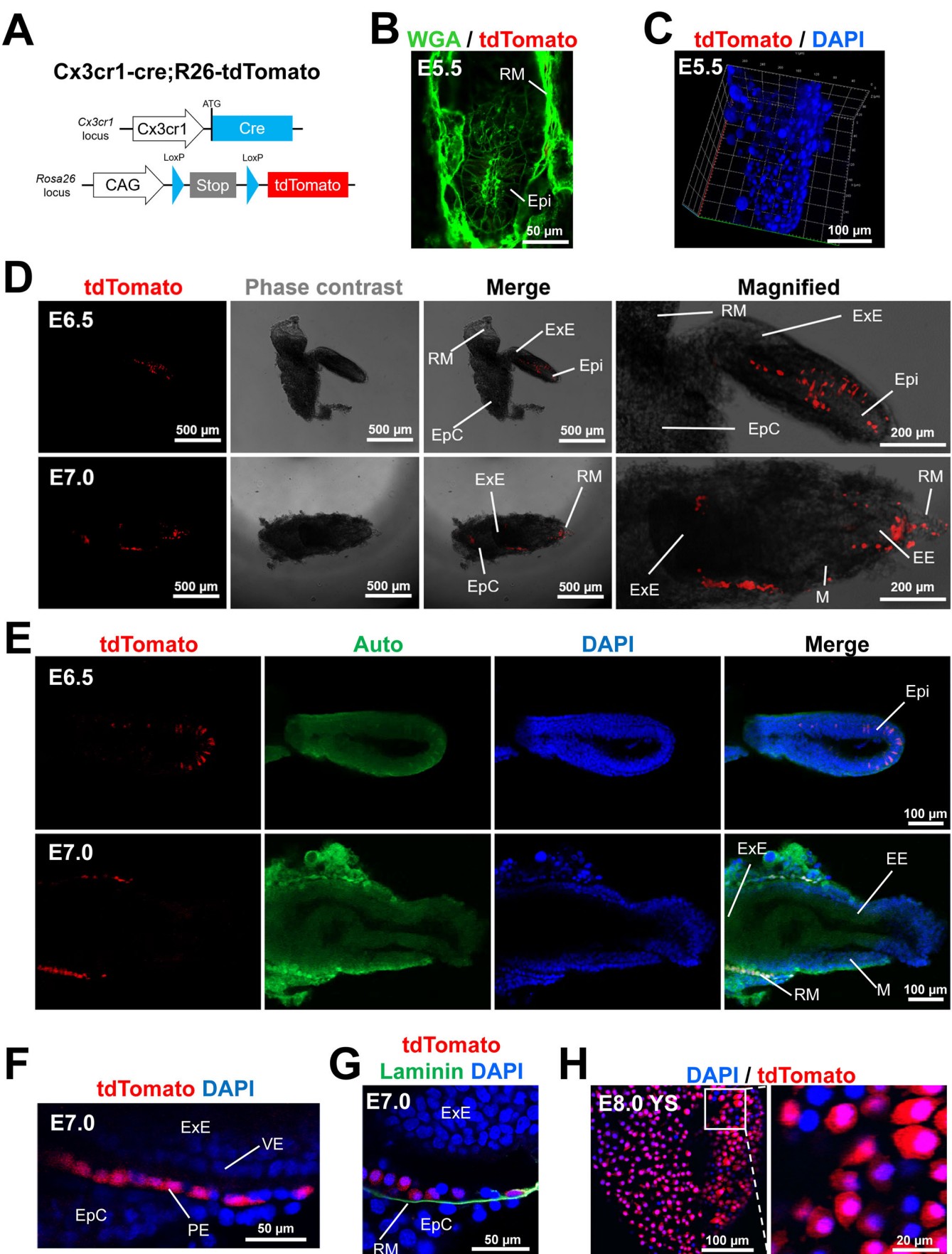

**Figure 1. Genetic lineage tracing of CX3CR1⁺ cells during early mouse development.**

(A) A schematic showing the genotype of embryos obtained by crossbreeding of *Cx3cr1-cre* mice and *R26-tdTomato* mice (*Cx3cr1-cre;R26-tdTomato*). The generated mice enable permanent labeling of CX3CR1⁺ cells and their progeny with tdTomato, a red fluorescent protein. (B) A representative confocal microscopic image of the E5.5 embryo stained for WGA (green fluorescence). (C) A 3D-reconstructed image of the E5.5 embryo stained for DAPI. (D) Representative phase contrast and fluorescent microscopic images of the embryo at E6.5 and E7.0. Epi epiblast, ExE extraembryonic ectoderm, EE embryonic ectoderm, RM Reichert's membrane, M mesoderm, EpC ectoplacental cone. (E) Representative confocal microscopic images of the embryos at E6.5 and E7.0. Auto autofluorescence. DAPI (blue). (F) A magnified confocal microscopic image of the embryo at E7.0. VE visceral endoderm, PE parietal endoderm. (G) A representative confocal microscopic image of the embryo at E7.0 stained for laminin. DAPI (blue). (H) Confocal microscopic images of the YS membrane at E8.0 after DAPI staining. All images are representative of three individual embryos. Source data are available online for this figure.

CX3CR1⁺ cells before E8.25 and their commitment other than to macrophages have not been investigated. Understanding the origin and fate of CX3CR1⁺ cells would help determine whether they can be harnessed for cardiac regeneration.

Accordingly, in this study, we investigated the origin and fate of CX3CR1⁺ cells. We found that CX3CR1⁺ cells emerge from a subset of epiblasts at E6.5. They prenatally contribute to the generation of cardiomyocytes (CMs) and endothelial cells (ECs) in addition to macrophages, through both de novo differentiation and fusion with preexisting CMs and ECs, respectively. Furthermore, CX3CR1⁺ cells differentiated from mouse embryonic stem cells (mESCs) in culture produced CMs, ECs, and macrophages in vitro, ex vivo and in vivo. Single-cell RNA sequencing (scRNA-seq) showed that *Cx3cr1⁺* cells represent an intermediate cell population differentiating into mesoderm from embryonic stem cells (ESCs). Together, this study demonstrated that E6.5 CX3CR1⁺ cells are multipotent progenitor cells first emerging from the epiblast, and they play an important role in cardiac development by differentiating into CMs, ECs, as well as macrophages.

## Results

### CX3CR1⁺ cells arise from a subset of epiblasts at E6.5 and differentiate into the parietal endoderm at E7.0

Studies reported that CX3CR1⁺ cells are present in the YS at E8.5–9.5 and in most fetal organs from E10.5 (Mass et al, 2016; Stremmel et al, 2018). However, it remained unknown whether CX3CR1⁺ cells could be present before E8.5. To address this issue, we cross-bred *Cx3cr1-cre* and *R26-tdTomato* mice and examined the embryos (Fig. 1A). This double knock-in system enables permanent genetic labeling of both CX3CR1⁺ cells and all their progeny with tdTomato, a red fluorescent protein. In this study, we defined these cells as *Cx3cr1* lineage cells. The embryos at E5.5 did not show tdTomato⁺ cells (Fig. 1B,C). We observed that tdTomato⁺ cells first appeared at E6.5 (Fig. 1D,E). The tdTomato signals were detected in a subset of epiblasts (Epi) but not in other parts such as extraembryonic ectoderm (ExE), ectoplacental cone (EpC), and Reichert's membrane (RM) at E6.5 (Fig. 1D,E, the upper lane).

To confirm this observation, we analyzed previously published data (Wen, Zeng et al, 2017). Wen et al collected embryos at E5.5–6.5 and performed scRNA-seq. We selected this dataset because our lineage tracing data indicated that *Cx3cr1* starts to be expressed between E5.5 and E6.5 (Fig. 1A–E). tSNE plot showed clearly three different major cell clusters that represent epiblasts (Epi), extraembryonic ectoderm (ExE), and mesoderm (ME)

(Fig. S1A,B). Our analysis showed that 10.6% of epiblasts express *Cx3cr1* (Fig. S1A). The tSNE plot showed that the majority of *Cx3cr1⁺* cells (82.4%, 14 out of 17 *Cx3cr1⁺* cells) were epiblasts. In addition, *Cx3cr1⁺* cells highly expressed *Nanog* compared to *Cx3cr1⁻* cells and epiblasts (Fig. S1C). For other pluripotency markers such as *Pou5f1, Sox2, Dnmt3b, Zfp43, Gdf3*, and *Dppa5a*, *Cx3cr1⁺* cells showed similar expression levels compared to those of epiblasts. This result is consistent with our observation that *Cx3cr1* was expressed in a subset of epiblasts at between E5.5 and E6.5.

At E7.0, the tdTomato signals were detected in RM but not in other parts, including Epi, ExE, and EpC (Fig. 1D,E, the lower lane). Magnified images further showed that tdTomato⁺ cells were localized in the parietal endoderm (PE) that is aligned between visceral endoderm (VE) and EpC (Fig. 1F). The presence of CXCR1⁺ cells in the PE was further confirmed by immunostaining with laminin, which is produced by PE cells (Cooper, Kurkinen et al, 1981) (Fig. 1G). The PE is known to give rise to the outer layer of the YS (Barbacci, Reber et al, 1999). At E8.0, tdTomato⁺ cells were detected in the YS (Fig. 1H). These results suggest that CX3CR1⁺ cells emerge from a subset of epiblasts at E6.5 and subsequently contribute to the PE, and later to the YS.

### CX3CR1⁺ cells and their progeny prenatally contribute to cardiomyocytes that persist into adulthood

A prior study reported that CX3CR1⁺ cells populate the early embryonic heart as early as E9.5 (Epelman et al, 2014). To determine the fate of CX3CR1⁺ cells and their progeny in the developing heart, we examined the *Cx3cr1-cre;R26-tdTomato* embryo at E9.5. Immunostaining of E9.5 embryos for TNNT2 showed that tdTomato⁺ cells colonized embryonic myocardium (Fig. 2A). A subset of tdTomato⁺ cells did not express TNNT2, suggesting that they are non-CMs (Fig. 2B,C). However, another subset of tdTomato⁺ cells expressed TNNT2 (Fig. 2C–F, arrow). These results suggest that CX3CR1⁺ cells and their progeny contribute to both CMs and non-CMs in the early developing heart as early as E9.5.

To further determine the contribution of CX3CR1⁺ cells and their progeny to postnatal CMs, heart tissues of adult *Cx3cr1-cre;R26-tdTomato* mice were harvested at 6 months postnatally (Fig. 3A–C). Immunostaining for ACTN2 demonstrated that tdTomato⁺ CMs were present throughout the cardiac regions including epicardium, myocardium, and endocardium (Fig. 3A). To quantify the percentage of CMs derived from CX3CR1⁺ cells and their progeny, hearts were enzymatically digested to single-cell suspensions, and the cells were stained for TNNT2 and subjected to flow cytometry. The results showed that 13.5 ± 0.2% of TNNT2⁺

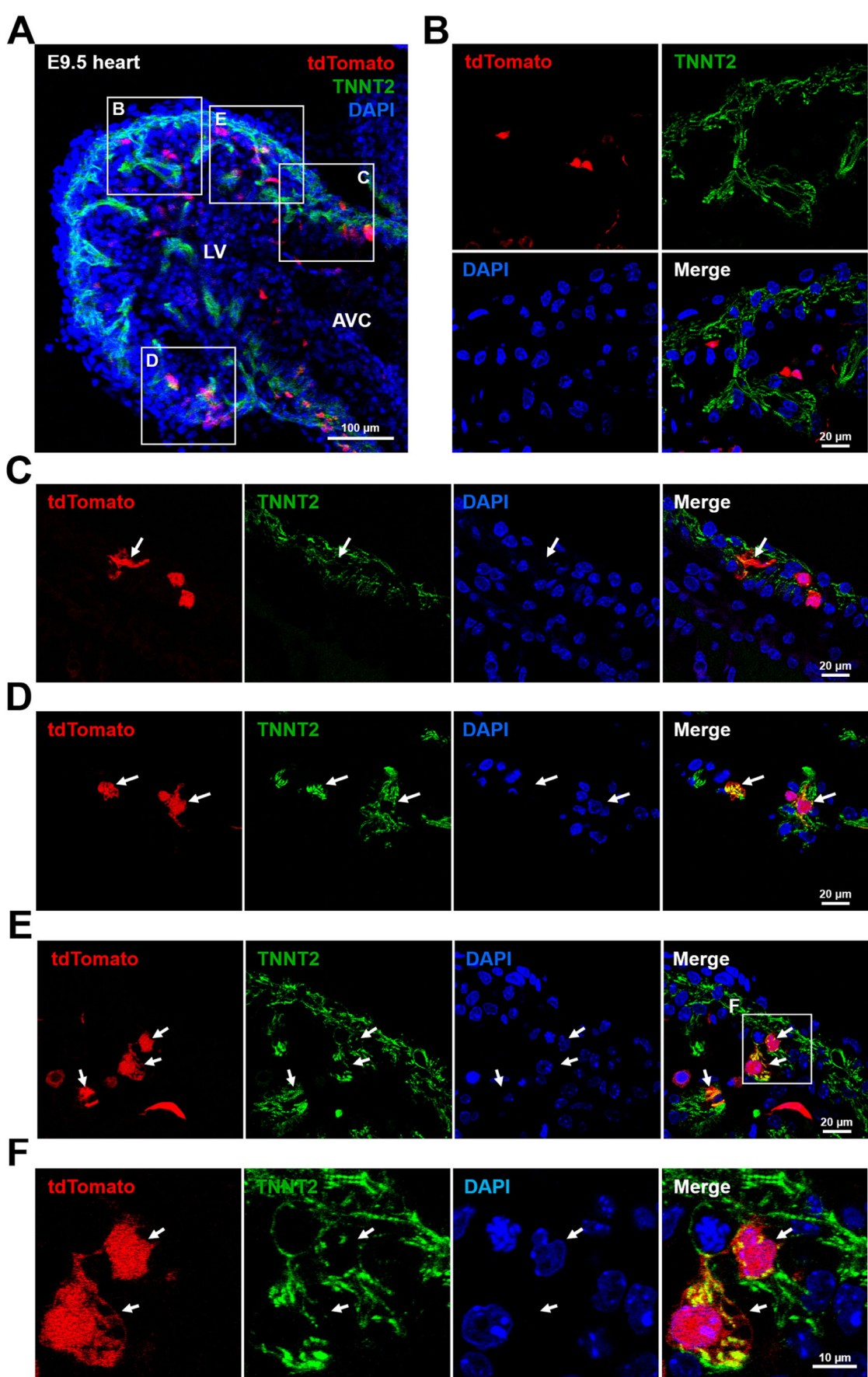

**Figure 2.  Contribution of CX3CR1⁺ cells and their progeny to embryonic CMs in the developing heart.**

(A) A representative confocal microscopic image of *Cx3cr1-cre;R26-tdTomato* mouse embryonic hearts at E9.5 in a cross-sectional view. The sectioned mouse embryos were immunostained for TNNT2 (green). DAPI (blue). (B–E) Magnified images of the boxed area in panel (A). Arrows indicate tdTomato⁺TNNT2⁺ cells, which represent CMs derived from CX3CR1⁺ cells and their progeny. (F) Magnified images of the boxed area in panel (E). lv left ventricle, avc atrioventricular canal. All images are representative of three individual embryos. Source data are available online for this figure.

CMs were positive for tdTomato (Fig. 3B,C). These data indicate that CMs derived from CX3CR1⁺ cells and their progeny persist into adulthood and comprise a significant portion of adult CMs.

To determine whether postnatal CX3CR1⁺ cells contribute to CMs, we crossed tamoxifen-inducible CX3CR1 driver mice (*Cx3cr1-creERT2*) with *R26-tdTomato* mice to generate *Cx3cr1-creERT2;R26-tdTomato* mice. Tamoxifen was administered to *Cx3cr1-creERT2;R26-tdTomato* mice at postnatal day 2 (P2) and mice were sacrificed 2 months later. Immunostaining of hearts for TNNT2 demonstrated that no tdTomato⁺ CMs were detected (Fig. 3D). Rather, tdTomato⁺ cells were localized in between CMs. These results showed that postnatal CX3CR1⁺ cells do not contribute to CMs. Together, these results indicate that CX3CR1⁺ cells and their progeny prenatally contribute to CMs that persist into adulthood.

## CX3CR1⁺ cells and their progeny generate CMs via both de novo differentiation and fusion with preexisting CMs

Previous reports showed that leukocytes can fuse with other somatic cells (Nygren, Jovinge et al, 2004). As CX3CR1 is also expressed on leukocytes (Imai et al, 1997), we wondered whether CX3CR1⁺ cells and their progeny contribute to CMs via fusion with preexisting cells or de novo differentiation. To address this issue, we generated *Cx3cr1-Cre;R26-mT/mG* mice. In these mice, cells that have never expressed the *Cx3cr1* gene are labeled with membrane-bound tdTomato (mT), and cells that have ever expressed the *Cx3cr1* gene are labeled with membrane-bound GFP (mG). Cells derived from a fusion between two different lineages express both mT and mG. In young adult mouse hearts (P30), all three types of CMs were observed: mT⁺ CMs, mG⁺ CMs (asterisks), and double-positive CMs (arrows) (Figs. 3E; S2A–C). Histological analyses showed that about 24% of CMs were generated by cell fusion (Fig. 3F). The same pattern was also observed in P1 neonatal hearts (Fig. S2D,E), suggesting that both de novo differentiation and fusion with preexisting cells occur during the prenatal period. To distinguish fusion-derived CMs from immune cells engulfing CMs, we immunostained the heart for CD45 (Fig. S2F). All CD45⁺ cells were small and round, suggesting that they are naïve immune cells. This means that immune cells engulfing CMs did not affect the calculation of the fusion rate. Together, these data demonstrate that CX3CR1⁺ cells or their progeny contribute to CMs through both de novo differentiation and fusion with preexisting CMs.

## CX3CR1⁺ cells and their progeny prenatally contribute to endothelial cells that persist into adulthood via both de novo differentiation and fusion with preexisting ECs

Since tdTomato signals were observed in non-CMs (Fig. 2), we also characterized the contribution of CX3CR1⁺ cells and their progeny

to cardiac ECs using *Cx3cr1-cre;R26-tdTomato* adult (6-month-old) mouse hearts. To stain endothelial cells (ECs), the adult mouse hearts were perfused with fluorescein-labeled BSL1. Confocal microscopic examination showed that a subset of tdTomato⁺ cells were stained positive for BSL1 (arrows) in both longitudinal (Fig. 4A) and cross-sectional views (Fig. 4B). To quantify ECs originated from CX3CR1⁺ cells and their progeny, BSL1-perfused hearts were enzymatically digested and subjected to flow cytometry (Fig. 4C). The results showed that among the cardiac ECs (fluorescein-BSL1⁺), around 31% were positive for tdTomato, suggesting significant contribution of CX3CR1⁺ cells and their progeny to cardiac vasculature. We further examined whether postnatal CX3CR1⁺ cells contribute to cardiac ECs. To this end, tamoxifen was administered to *Cx3cr1-creERT2;R26-tdTomato* mice at P2 and the mice were perfused with fluorescein-BSL1 at 2 months. Confocal microscopic examination showed that none of the tdTomato⁺ cells were colocalized with BSL1⁺ ECs (Fig. 4D), indicating no commitment of postnatal CX3CR1⁺ cells to ECs after birth. Together, these data indicate that only prenatal CX3CR1⁺ cells and their progeny contribute to cardiac ECs.

To explore how CX3CR1⁺ cells contribute to ECs, the hearts from *Cx3cr1-cre;R26-mT/mG* young adult mice (P30) were stained for PECAM1, an EC marker. Similar to CMs, three types of ECs were observed: mT⁺ ECs, mG⁺ ECs (asterisk), and double-positive ECs (arrow) (Fig. 4E). Histological analyses showed that ~18% of ECs derived from CX3CR1⁺ cells and their progeny were generated by cell fusion (Fig. 4F). Taken together, CX3CR1⁺ cells and their progeny prenatally contributed to ECs that persist into adulthood through both de novo differentiation and fusion with preexisting ECs.

## E6.5 CX3CR1⁺ cells contribute to CMs and ECs

Although our data suggest that CX3CR1⁺ cells and their progeny emerge from epiblasts and generate CMs and ECs, it is still possible that CX3CR1 could be transiently expressed in differentiating CMs and ECs. To rule out this possibility, we genetically pulse-labeled CX3CR1⁺ cells by injecting tamoxifen into *Cx3cr1-creERT2;R26-tdTomato* mice at E6.5 (Fig. 5A,B). We harvested the embryo at E10.5 and found that tdTomato⁺ cells were localized to different parts of the embryos, such as the heart, head, and back (Fig. 5C,D). Immunostaining showed that tdTomato⁺ cells expressed ACTN2 and PECAM1 (Fig. 5E,F). We did not observe tdTomato⁺ macrophages in the heart at this stage (Fig. 5G), while tdTomato⁺ cells contributed to ECs and macrophages (expressing CD68, a pan-macrophage marker) in the brain (Fig. 5H,I). When tamoxifen was injected at E8.5, we found that tdTomato⁺ cells expressed CD68 in the heart (Fig. 5J). These results suggest that E8.5 CX3CR1⁺ cells (non-epiblasts) contribute to cardiac macrophages, whereas E6.5 CX3CR1⁺ cells (epiblasts) contribute to microglia. Together, these data indicate that E6.5 CX3CR1⁺ cells directly generate CMs and ECs in the developing heart.

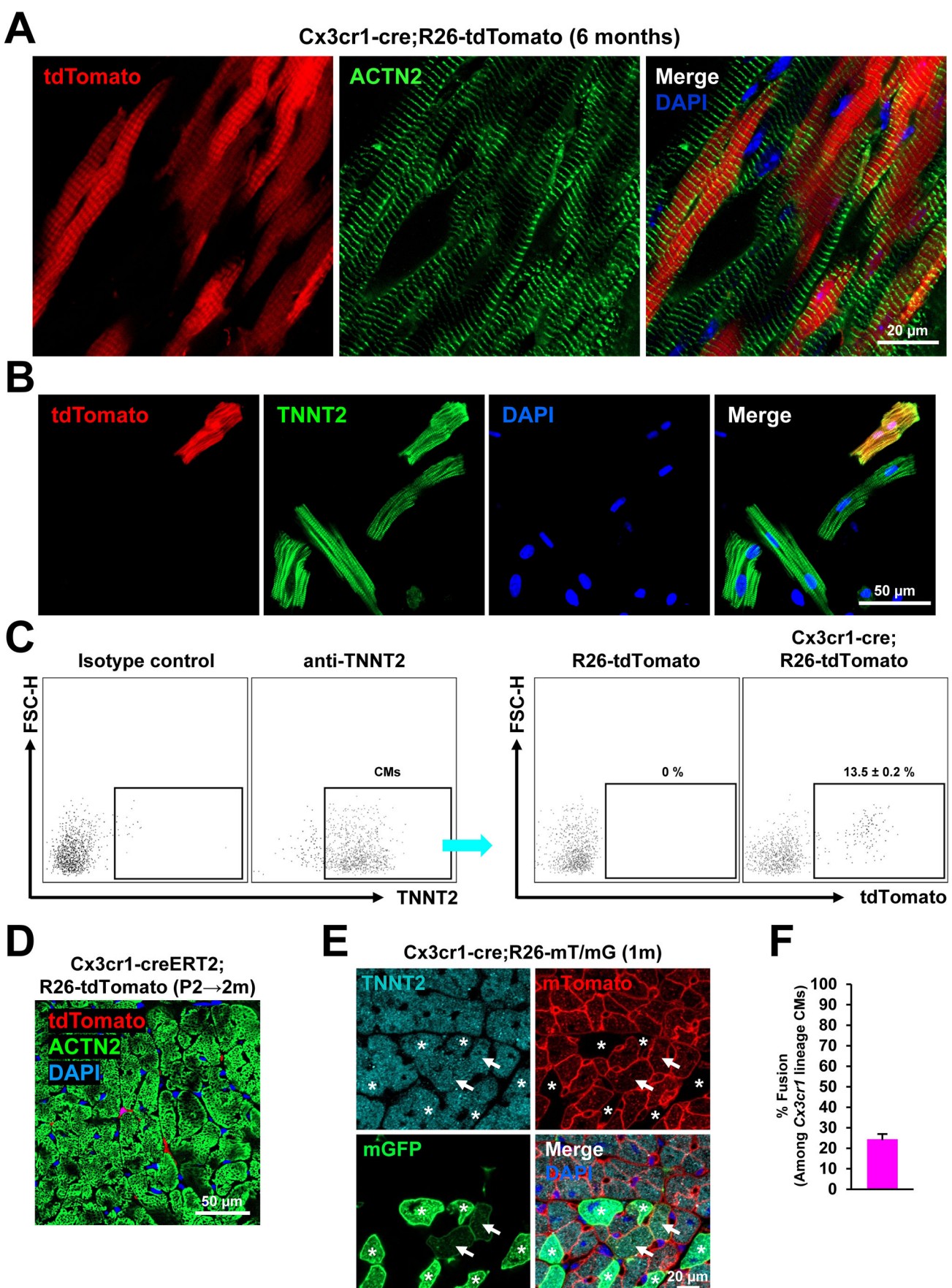

◄ **Figure 3.  Prenatal contribution of CX3CR1⁺ cells and their progeny to CMs that persist into adulthood via both de novo differentiation and fusion with preexisting CMs.**

(A) Representative confocal microscopic images of the adult heart (6 months old) of a *Cx3cr1-cre;R26-tdTomato* mouse. Heart tissues were sectioned and stained for ACTN2 (green). tdTomato (red). DAPI (blue). (B) Confocal microscopic images of the CMs dissociated from a *Cx3cr1-cre;R26-tdTomato* mouse (6 months old) heart after staining for TNNT2 and DAPI. (C) Flow cytometric plots for the dissociated CMs used in panel B ($n = 8$). TNNT2⁺ CM population was further gated to analyze the percentage of tdTomato⁺ cells. R26-tdTomato mice of the same age were used as a negative control. Cell population percentages were presented as mean ± standard error of the mean (s.e.m). (D) A representative confocal microscopic image of the *Cx3cr1-creERT2;R26-tdTomato* heart stained for TNNT2. P2 neonatal mice were given tamoxifen via a single subcutaneous injection, and hearts were harvested after 2 months. A confocal microscopic image of a one-month-old *Cx3cr1-cre;R26-mT/mG* heart and the quantification of the percentage of fusion-derived CMs. Asterisks indicate de novo CMs derived from *Cx3cr1* lineage cells and arrows indicate CMs derived from fusion between *Cx3cr1* lineage cells and preexisting CMs. DAPI (blue). The error bar is presented as mean ± s.e.m. (E, F) Quantification of the percentage of fusion-derived CMs. Five mice were examined. Representative images from nine different sections per mouse were assessed. In each mouse, the number of double-positive cells was summed and divided by the total number of mGFP-positive cells to calculate the fusion rate for each mouse. The error bar is presented as mean ± s.e.m. Source data are available online for this figure.

## Mouse lines used for genetic lineage tracing specifically label CX3CR1⁺ cells without leakiness

Previous reports raised concerns about the specificity and leakiness of the *Cre-loxP* system (Abram, Roberge et al, 2014; Bittner-Eddy, Fischer et al, 2019; Donocoff, Teteloshvili et al, 2020; McCubbrey, Allison et al, 2017; Stifter and Greter, 2020). To examine the specificity and leakiness of *Cx3cr1-Cre* and *Cx3cr1-CreERT2* mice, we collected adult hearts and immunostained for CRE and CX3CR1 (Fig. S3A–D). Comprehensive confocal microscopic analysis of the heart showed that expression of CRE (Fig. S3A,B) and CREERT2 (Fig. S3C,D) proteins was confined only to CX3CR1⁺ cells. We did not observe CX3CR1⁻ cells that express CRE or CREERT2. These results indicate that the expression of CRE and CREERT2 is specific to CX3CR1⁺ cells without leakage in CX3CR1⁻ cells. Furthermore, we examined the leakiness of the reporter mouse lines. We did not detect leaked signals in either *R26-mT/mG* (Fig. S4A) or *R26-tdTomato mice* (Fig. S4B). These data suggest that the mouse lines used for genetic lineage tracing specifically label CX3CR1⁺ cells without leakiness.

## CX3CR1 temporally marks a subset of epiblasts but does not mark hemogenic cells

Genetic lineage tracing data suggest that CX3CR1⁺ cells have the capacity to produce both endothelial cells and macrophages (Figs. 4 and 5). However, it is unclear whether CX3CR1 is expressed in hemogenic endothelium or angioblasts. Reports showed that developing endocardium can generate hematopoietic cells (Nakano, Liu et al, 2013; Zamir, Singh et al, 2017). To examine expression of CX3CR1 in the developing endocardium, we performed immunostaining with an anti-CX3CR1 antibody. We used PECAM1 as an endothelium marker and CD41 as an early hematopoietic progenitor marker (Mitjavila-Garcia, Cailleret et al, 2002; Zovein, Hofmann et al, 2008). In the developing heart, CX3CR1 was not colocalized with PECAM1⁺ endocardial cells or CD41⁺ hemogenic cells (Fig. S5A). As a positive control, we stained the developing heart for CD68 and detected its colocalization with CX3CR1 in the epicardium area. The liver bud is another embryonic organ for active hematopoiesis (Gordillo, Evans et al, 2015). Thus, we examined whether CX3CR1 is expressed in the endothelium or early hematopoietic progenitors in the developing liver (Fig. S5B). Confocal microscopic analyses showed that CX3CR1 was not detected on PECAM1⁺ endothelial cells nor on CD41⁺ hemogenic

cells in the liver bud. As a positive control, we stained the developing liver for CD68 and detected its colocalization with CX3CR1. Together, these results demonstrate that CX3CR1⁺ cells are not derived from the hemogenic population in the heart or in the liver bud.

Furthermore, we checked the expression pattern of CX3CR1 at different embryonic stages using an anti-CX3CR1 antibody. We found that CX3CR1 was transiently expressed at E6.5 and turned off at E7.5 (Fig. S5C). At E8.5–9.5, we observed CX3CR1⁺ cells in the developing embryo, which is consistent with previous reports (Epelman et al, 2014; Schulz et al, 2012). These data together with our genetic lineage tracing data (Fig. 1) show that CX3CR1 temporally marks a subset of epiblasts between E5.5 and E6.5.

## Mouse ESC-derived CX3CR1⁺ cells generate CMs and ECs in vitro

To determine whether this multipotency of prenatal CX3CR1⁺ cells can be employed for regenerative medicine, we attempted to differentiate mESCs into CX3CR1⁺ cells. Because CMs and ECs are derived from the mesoderm, we questioned whether CX3CR1 is expressed during mesodermal differentiation of mESCs. To differentiate mESCs into mesodermal lineages, we utilized an OP9 (bone marrow stromal cell line) coculture system (Lynch, Gasson et al, 2011; Vodyanik, Bork et al, 2005). mESCs (J1) were cultured on a monolayer of OP9 cells, and CX3CR1 expression was examined daily via flow cytometry. The results showed that the CX3CR1⁺ population increased until day 5 (D5) (Fig. S6A) and gradually decreased afterwards. To gain insight into the identity of these differentiated CX3CR1⁺ cells, we checked expression of colony-stimulating factor 1 receptor (CSF1R) and CD166 via flow cytometry at D5. CSF1R is known as a marker for erythro-myeloid progenitors derived from the YS (Gomez Perdiguero, Klapproth et al, 2015), and CD166 was reported to be expressed on cardiovascular progenitor cells that can produce CMs and ECs (Gessert, Maurus et al, 2008; Hirata, Murakami et al, 2006; Murakami, Hirata et al, 2007). Flow cytometric data showed that mESC-CX3CR1⁺ cells minimally expressed CSF1R, but expressed CD166 at about 80% (Fig. S6B). This high expression of CD166 in mESC-CX3CR1⁺ cells suggests their differentiation potential toward CMs and ECs.

Since CX3CR1 is known as a marker for myeloid cells (Gordon and Taylor, 2005), we checked whether mESC-CX3CR1⁺ cells express myeloid markers such as CD11b, CD14, F4/80, and Gr-1 via flow

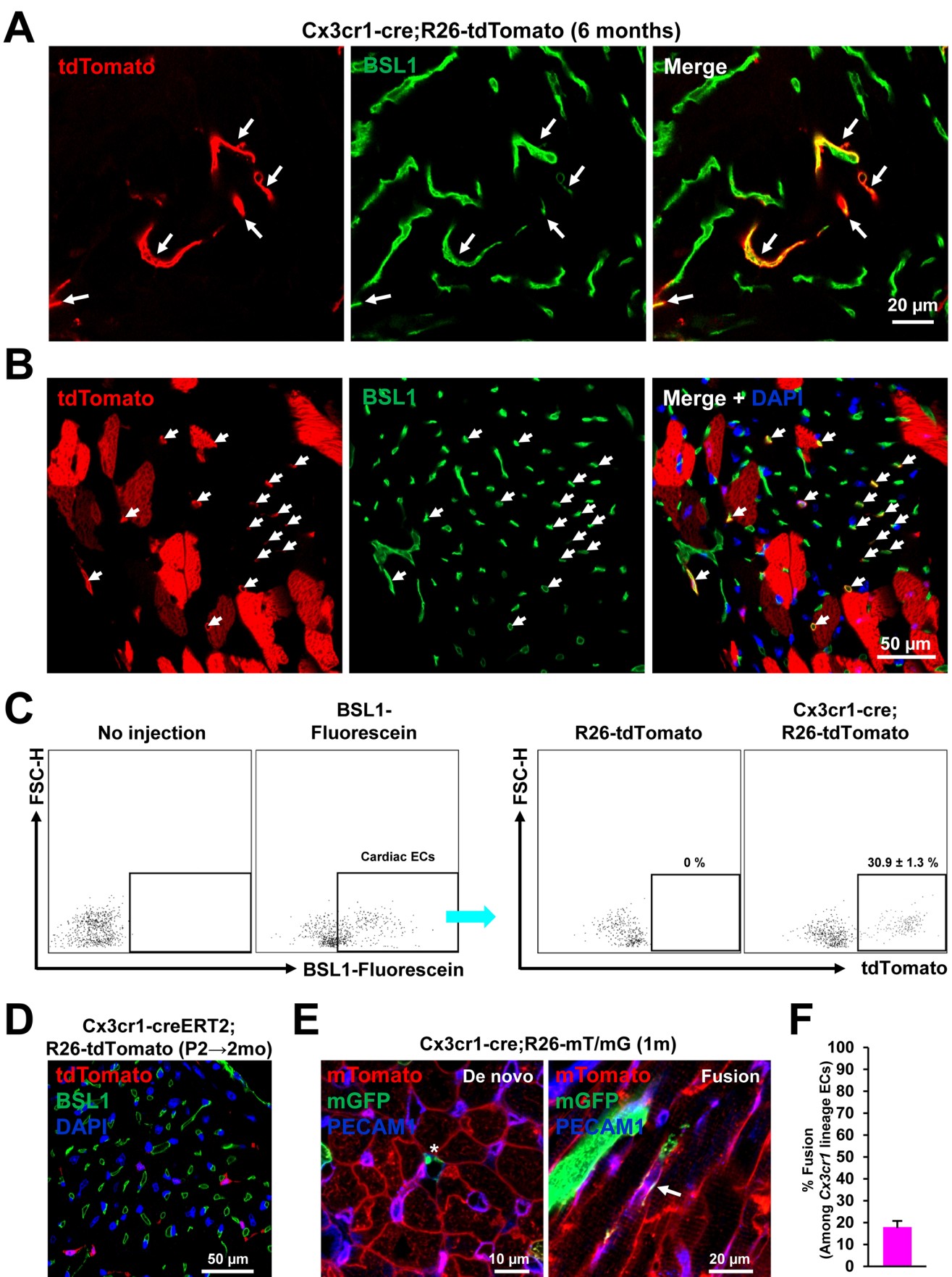

◄ **Figure 4. Prenatal contribution of CX3CR1⁺ cells and their progeny to cardiac ECs that persist into adulthood via both de novo and fusion with preexisting ECs.**

(A, B) Representative confocal microscopic images of the *Cx3cr1-cre;R26-tdTomato* heart in a longitudinal section (A) and in a cross section (B). Adult (6 months old) mouse hearts were perfused with fluorescein-labeled BSL1 to visualize blood vessels. Arrows indicate cells with both tdTomato and BSL1 signals. (C) BSL1-perfused hearts were enzymatically digested and subjected to flow cytometry, showing the percentage of *Cx3cr1* lineage ECs ($n = 3$). Cell population percentages were presented as mean ± standard error of the mean (s.e.m.). (D) A representative confocal microscopic image of the *Cx3cr1-creERT2;R26-tdTomato* heart perfused with BSL1. P2 neonatal mice were given tamoxifen via a single subcutaneous injection, and hearts were harvested after 2 months. (E) Representative confocal microscopic images of the 1-month-old *Cx3cr1-cre;R26-mT/mG* hearts stained for PECAM1. An asterisk indicates de novo ECs derived from CX3CR1⁺ cells and their progeny and an arrow indicates ECs derived from cell fusion. DAPI (blue). (F) Quantification of the percentage of fusion-derived ECs. Three mice were examined. Representative images from seven different sections per mouse were assessed. In each mouse, the number of double-positive cells was summed and divided by the total number of mGFP-positive cells to calculate the fusion rate for each mouse. The error bar is presented as mean ± standard error of the mean (s.e.m). Source data are available online for this figure.

cytometry. The results showed that mESC-CX3CR1⁺ cells did not express these markers (Fig. S6C). We also checked the expression of genes related to myeloid differentiation such as Tal1 (a transcription factor required for differentiation of mesoderm to hematopoietic stem cells), Cebpa (a transcription factor required for differentiation of hematopoietic stem cells to granulocyte-monocyte progenitors), and Irf8 (a transcription factor required for differentiation of monoblasts to macrophages) (Rosenbauer and Tenen, 2007) via qRT-PCR. We used J1 mouse ESCs as a negative control and adult mouse bone marrow (BM) as a positive control. We found that mESC-CX3CR1⁺ cells minimally express these markers without significant difference from J1 mESCs (Fig. S6D). These results suggest that mESC-CX3CR1⁺ cells are not myeloid lineage cells.

Since our genetic lineage tracing data suggest that CX3CR1⁺ cells contribute to CMs and ECs (Fig. 5), we checked whether mESC-CX3CR1⁺ cells express markers for cardiac progenitors such as *Nkx2-5, Gata4, Isl1, Tbx5*, and *Mef2c*. We used J1 mouse ESCs as a negative control and E10.5 mouse hearts as a positive control. We found that mESC-CX3CR1⁺ cells showed higher expression of *Nkx2-5* and *Gata4* compared to J1 mESCs and CX3CR1⁻ cells (Fig. S6E). For *Isl1, Tbx5* and *Mef2c*, there was no significant difference between CX3CR1⁺ cells and CX3CR1⁻ cells. These data suggest that mESC-CX3CR1⁺ cells have partial characteristics of cardiac progenitors and are more similar to the first heart progenitor (*Nkx2-5*⁺) than the second heart progenitor (*Isl1*⁺).

We then sought to determine whether CX3CR1⁺ cells can be further differentiated into the cardiac lineage. For cardiac differentiation, CX3CR1⁺ cells were sorted from differentiating mESCs at D5 via MACS and cultured with BMP4, Wnt inhibitor CHIR99021, and VEGFA for another 8 days. BMP4 and CHIR99021 are effective inducers of mesodermal differentiation (Burridge, Keller et al, 2012), and VEGFA is known to induce both cardiac and vascular specification (Giacomelli, Bellin et al, 2017). Thereafter, we treated the cells with L-ascorbic acid for another 2 to 7 days to promote proliferation of cardiac progenitor cells (Cao, Liu et al, 2012) (Fig. 6A, upper). At D15-20, we observed expression of CM markers (TNNT2 and MYH6) (Figs. 6B and S7A,B). TNNT2⁺ cells displayed morphological variability, including round, spindle, and polygonal shapes (Fig. S7A). Clusters of MYH6⁺ cells were also detected (Fig. S7B). Round morphology and less organized sarcomeric structures suggested immature CM characteristics. Furthermore, we checked EC markers (BSL1 and CDH5) and found BSL1⁺ cells and CDH5⁺ cells, which formed vessel-like structures in the same culture conditions (Fig. S7C). To further assess the capacity of mESC-CX3CR1⁺ cells to differentiate into ECs, we cultured MACS-isolated CX3CR1⁺ cells (D5) in endothelial growth medium (EGM2) (Fig. 6A, lower). We loaded cells on

top of Matrigel to provide an in vivo-like 3D environment. Then, we monitored cellular behavior daily until D25 under the microscope. At D6, sprouting cells were observed, and they continued to grow with longer branches (Fig. S7D). From D11, cells formed networks, which became larger until D25, when we stopped monitoring. During culture, cells penetrated into the Matrigel, reaching the bottom of the culture dishes, and spread out, forming networks (Fig. S7D, far right). We stained these cells for EC markers BSL1 and KDR, and found that most of the cells were positive (Figs. 6C; S7E). Together, these data indicate that under specific culture conditions, mESCs-CX3CR1⁺ cells can differentiate into CMs and ECs.

## Mouse ESC-derived CX3CR1⁺ cells differentiate into CMs and ECs in the fetal mouse heart ex vivo

We then determined whether these mESC-CX3CR1⁺ cells can give rise to CMs and ECs in the ex vivo heart. We differentiated J1 mESCs into CX3CR1⁺ cells via coculture with OP9 cells, sorted them via FACS at D5, and labeled them with the red fluorescent dye CM-DiI for cellular tracking under histological examination. These mESC-CX3CR1⁺ cells were cocultured with E15.5 fetal mouse hearts ex vivo for 20 days (Fig. 6D). Briefly, hearts were embedded within Matrigel in the culture dish (Dyer and Patterson, 2013). Then, we loaded cells plus culture medium on top of Matrigel. Over 20 days, we examined the fate of live DiI⁺ (CX3CR1⁺) cells in the hearts at predetermined time points under fluorescent microscopy. At the early phase (D1-2), DiI⁺ cells were observed on the surface of the heart, indicating their migration toward the fetal heart (Fig. S8A). After D4, DiI⁺ cells were localized at various places deeper in the fetal heart, suggesting their continuous migration into the heart (Fig. S8B). Immunostaining of the sectioned hearts showed that DiI⁺ cells expressed markers for CMs (ACTN2) and ECs (PECAM1) at D7 (Figs. 6E,F; S8C,D). Even outside of the fetal mouse heart, we observed that DiI⁺ cells sprouted and formed vessel-like networks within Matrigel, suggesting their robust vessel-forming activities (Fig. S8E). These data indicate that CX3CR1⁺ cells contribute to CMs and ECs in the fetal heart ex vivo.

## Mouse ESC-derived CX3CR1⁺ cells differentiate into CMs and ECs in the adult mouse heart in vivo

To further determine the capacity of mESC-CX3CR1⁺ cells to generate CMs and ECs in vivo, FACS-isolated mESC-CX3CR1⁺ cells were labeled with DiI and injected into the adult (3-month-old) mouse heart (Fig. 6G). To increase the survival and retention of transplanted cells, we encapsulated the cells within an injectable

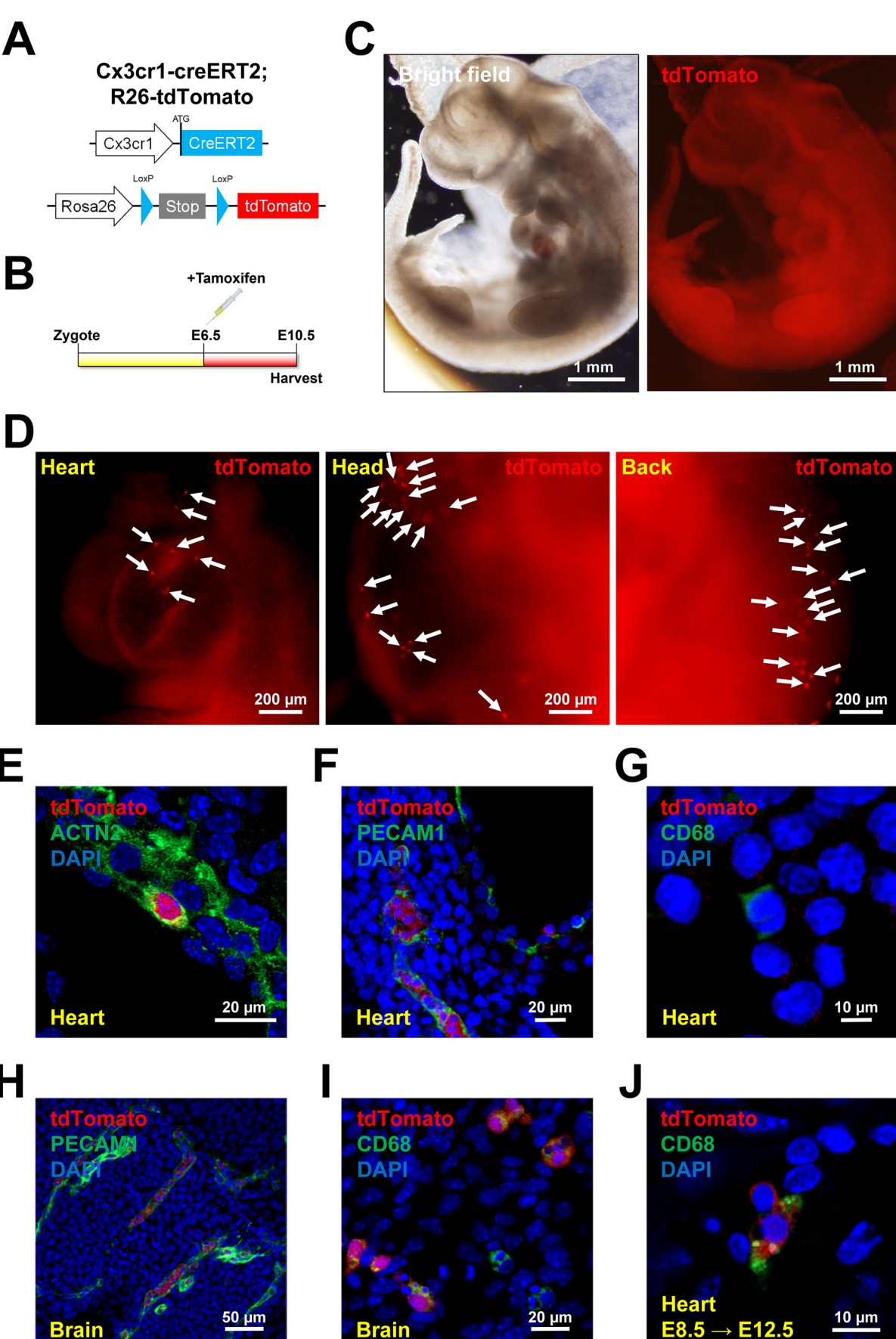

**Figure 5. Generation of CMs, ECs, and macrophages by E6.5 CX3CR1⁺ cells during embryonic development.**

(A) A schematic showing the genotype of mice obtained by crossbreeding *Cx3cr1-creERT2* and *R26-tdTomato* mice (*Cx3cr1-creERT2;R26-tdTomato*) used in this figure. (B) The experimental timeline for lineage tracing of CX3CR1⁺ cells for panels C-I. Pregnant mice with E6.5 embryos were given tamoxifen via a single peritoneal injection, and embryos were harvested at E10.5. (C, D) Representative bright field and fluorescent microscopic images of the embryo at E10.5 taken at low (C) and high (D) magnifications. Arrows indicate cells positive for tdTomato signal. (E–J) Representative confocal microscopic images of the heart and brain stained for ACTN2 (E), PECAM1 (F, H) and CD68 (G, I, J) as indicated. DAPI (blue). For panel (J), tamoxifen was administered at E8.5 and embryos were harvested at E12.5. All images are representative of three individual embryos. Source data are available online for this figure.

nanomatrix (PA-RGDS) before injection (Ban, Park et al, 2014). After 10 days, we collected hearts and performed histological analysis. We found that a significant number of cells were engrafted in the adult heart (Fig. 6H). Heart sections were immunostained for ACTN2 and PECAM1 (Fig. 6I–L). We found that a subset of injected cells (DiI⁺) was stained for either ACTN2 (Fig. 6J) or PECAM1 (Fig. 6L). These data suggest that mESC-CX3CR1⁺ cells can differentiate into CMs and ECs in the adult heart in vivo.

### CX3CR1⁺ cells contribute to macrophages in vivo, in vitro, and ex vivo

Previous reports showed that CX3CR1⁺ cells can generate tissue macrophages, an essential component of organogenesis (Mass et al, 2016; Schulz et al, 2012; Yona et al, 2013). Thus, we also wanted to verify the contribution of CX3CR1⁺ cells to cardiac macrophages using the constitutive *Cx3cr1-cre* driver mouse line and mESC-CX3CR1⁺ cells. Immunostaining of the *Cx3cr1-cre;R26-tdTomato* adult (6-month-old) mouse hearts for CD68 showed that a subset of tdTomato⁺ cells expressed CD68 (Fig. S9A). We also cultured MACS-isolated mESC-CX3CR1⁺ cells in the presence of M-CSF for 5 days. At D10, we stained them for CD68, and found a large number of CD68⁺ cells (Fig. S9B). Then, FACS-isolated CX3CR1⁺ cells labeled with DiI were cocultured with fetal mouse hearts ex vivo. At D20, the heart sections were immunostained for CD68. Confocal microscopic examination showed that a subset of DiI⁺ cells were colocalized with CD68 (Fig. S9C). Finally, a subset of mESC-CX3CR1⁺ cells, when transplanted to adult mouse hearts, expressed CD68 (Fig. S9D). Together, these results suggest that CX3CR1⁺ cells can give rise to macrophages in vivo, in vitro, and ex vivo.

### *Cx3cr1⁺* cells represent an intermediate cell population differentiating into mesodermal cells from pluripotent stem cells

To characterize the heterogeneity and differentiation trajectory of mESC-CX3CR1⁺ cells at the single-cell level, we performed scRNA-seq (Figs. 7; S10). J1 mESCs were cocultured with OP9 cells and collected at Day 5 for scRNA-seq (Fig. 7A). Undifferentiated mESCs were also included as a control group. After filtering out low-quality cells (Fig. S10A), we performed principal component analysis (PCA) and Jackstraw analysis to identify significant principal components and to reduce dimensionality (Fig. S10B,C). Then, we utilized the uniform manifold approximation and projection (UMAP) algorithm to visualize 19,550 cells (Fig. 7B). We identified distinct cell clusters: pluripotent stem cells, mesodermal cells, smooth muscle cells, and cell clusters without significant expression of cell type-specific genes, which we named

"unknown lineages" (UL) 1–4. OP9 cells were removed from the UMAP, since they are not derived from mESCs. We annotated each cluster based on its expression of cell type-specific genes, and this annotation was confirmed by Enrichr (Chen, Tan et al, 2013) (Figs. 7C; S10D). Cell cycle analysis indicated that UL4 showed the lowest percentage of cells at the G1 phase among cell clusters differentiated from PSCs, suggesting that UL4 is the most proliferative cell cluster after PSCs (Fig. 7D). To identify cell clusters that express *Cx3cr1*, we calculated the percentage of *Cx3cr1⁺* cells, and UL4 showed the highest percentage (Fig. 7E). To examine differentiation trajectories, we performed diffusion map analysis (Haghverdi, Buettner et al, 2015). Three tips and three branches were detected, suggesting that mESCs were differentiated into three major different lineages (Fig. S10E). Further analysis showed that UL4 was aligned between PSCs and mesodermal cells (Fig. 7F,G). These results suggest that *Cx3cr1⁺* cells represent a unique intermediate cell population differentiating to the mesoderm from PSCs.

To analyze transcriptional signatures of *Cx3cr1⁺* cells, we selected *Cx3cr1⁺* cells from the UL4 cluster and examined their protein–protein interaction (PPI) network using the STRING database and was subsequently analyzed by the Cytoscape program (Fig. S10F). We found that the core network of *Cx3cr1⁺* cells comprises transcription factors including Zfp352, Zscan4, and Tcstv (Fig. S10G,H). Zfp352 activates developmental programs during embryogenesis (Mwalilino, Yamane et al, 2023). Zscan4 has multiple functions including derepression of heterochromatin, maintenance of telomere length, and genome stability in mESCs (Akiyama, Xin et al, 2015; Zalzman, Falco et al, 2010). Tcstv elongates telomeres of mESCs (Zhang, Dan et al, 2016). These reports suggest that *Cx3cr1⁺* cells have features of both differentiating cells and mESCs. This supports our notion that *Cx3cr1⁺* cells represent a transitional differentiating cell population exiting the pluripotent state, and their in vivo equivalent would be a subset of primed epiblasts emerging between E5.5 and E6.5.

## Discussion

This study identified CX3CR1⁺ cells at E6.5 as a multipotent progenitor population originating from a subset of epiblasts. This progenitor population contributes to CMs and ECs in the heart through both de novo differentiation and cell fusion during early embryonic development. Using genetic lineage tracing approaches, our study revealed that prenatal, but not postnatal, CX3CR1⁺ cells contribute to the generation of CMs and ECs, which persist into adulthood. Furthermore, CX3CR1⁺ cells differentiated from mESCs have the potential to differentiate into CMs, ECs and macrophages, as confirmed by in vitro, ex vivo and in vivo models. Our scRNA-

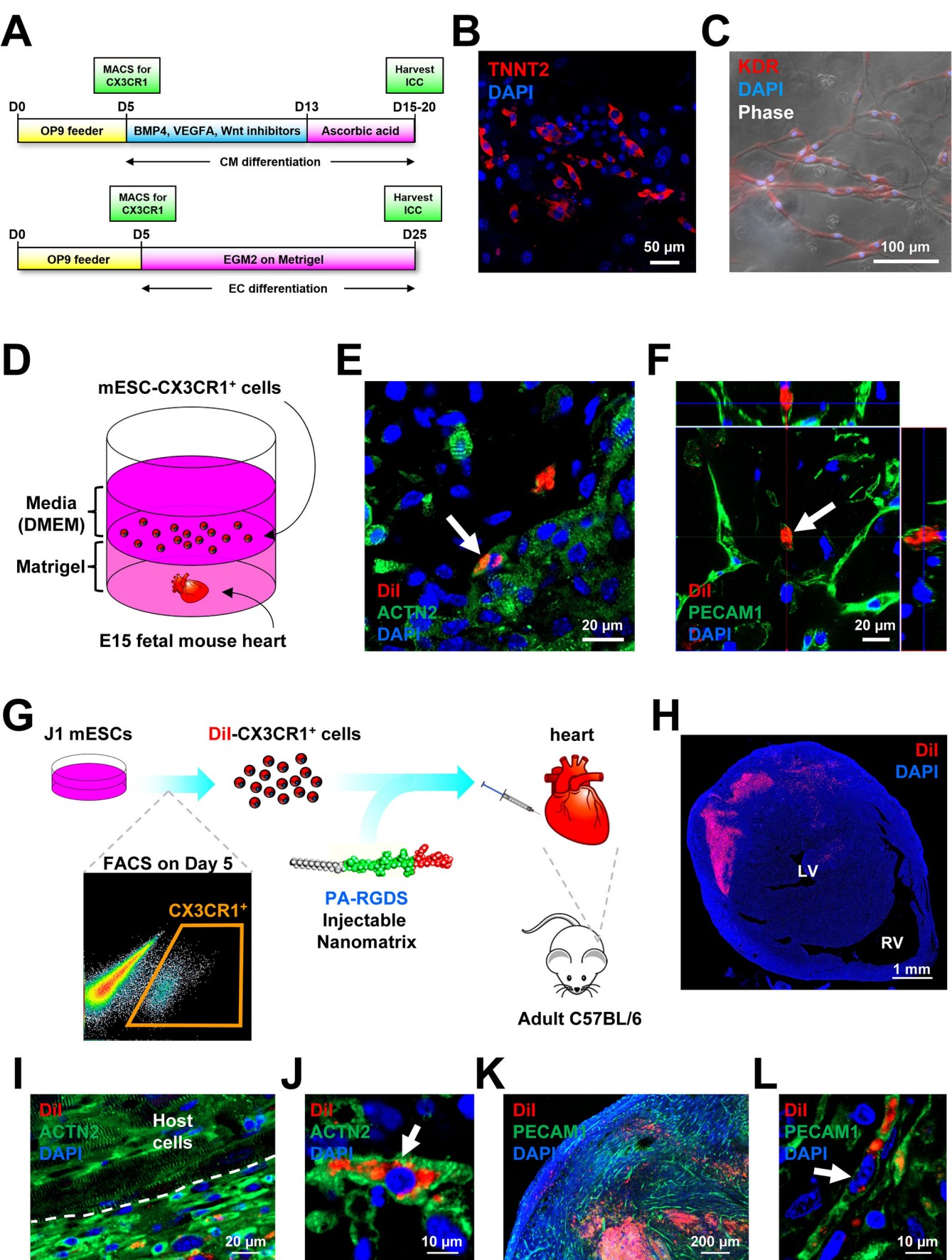

◄ **Figure 6. Generation of CX3CR1⁺ cells from differentiating mESCs and their characterization in vitro, ex vivo, and in vivo.**

(A) A schematic of experimental procedures used in panels B-C. mESCs were cocultured with OP9 cells and CX3CR1⁺ cells were sorted by MACS at D5. Cells were further cultured in cardiac (upper) or endothelial (lower) differentiation conditions. (B, C) Representative confocal microscopic images of cultured mESC-CX3CR1⁺ cells stained for TNNT2 at D15-20 (B) and KDR at D25 (C). DAPI (blue). (D) A schematic of experimental procedures used in panels E-F. FACS-isolated mESC-CX3CR1⁺ cells were labeled with DiI and loaded on top of the Matrigel containing E15.5 fetal mouse hearts (n = 12). (E, F). Representative confocal microscopic images of fetal mouse hearts stained for ACTN2 (E) and PECAM1 (F) at D7. Arrows indicate CMs (E) and ECs (F) contributed by DiI-labeled mESC-CX3CR1⁺ cells. DAPI (blue). (G) A schematic of experimental procedures used in panels H-L. FACS-isolated mESC-CX3CR1⁺ cells were labeled with DiI and injected into the adult mouse hearts together with injectable nanomatrix PA-RGDS (n = 3). (H) A representative confocal microscopic image of the heart at 10 days post-injection. LV left ventricle, RV right ventricle. (I–L) Representative confocal microscopic images of the heart stained for ACTN2 (I, J) and PECAM1 (K, L). Arrows indicate CMs (J) and ECs (L) contributed by mESC-CX3CR1⁺ cells. DAPI (blue). Source data are available online for this figure.

seq data further suggest that CX3CR1⁺ cells represent an intermediate population transitioning from ESCs into the mesoderm.

Our study provides insights into how epiblasts segregate into heterogeneous populations. Shortly before implantation, a cluster of 10–20 nascent epiblasts is localized between trophoblasts and hypoblasts at E3.5–4.5 (Batlle-Morera, Smith et al, 2008). These pre-implantation epiblasts are unspecialized and possess unrestricted lineage potential and self-renewal capacity (Gardner, 1998). Thus, they are termed "naive epiblasts" with ground-state pluripotency (Nichols and Smith, 2009). Shortly after implantation, epiblasts form a cup-shaped structure of a single-cell layer, known as the egg cylinder, at E5.5 (Kaufman, 1992). These post-implantation epiblasts are exposed to inductive stimuli such as FGFs, BMPs, and Wnts from extraembryonic tissues (Beddington and Robertson, 1999). Thus, they are termed "primed epiblasts" (Nichols and Smith, 2009). During the naive-to-primed pluripotency transition of the epiblast, significant changes in transcriptomic and epigenomic features occur (Davidson, Mason et al, 2015; Hackett and Surani, 2014). At E6.5, epiblasts begin to exit the pluripotency state and differentiate into the primitive streak, showing transcriptional heterogeneity (Mohammed, Hernando-Herraez et al, 2017). Our data showed that *Cx3cr1* is expressed between E5.5 and E6.5, suggesting that *Cx3cr1* marks a unique subset of primed epiblasts differentiating into the PE. However, it is still unclear whether expression of *Cx3cr1* is the cause or result of transcriptional heterogeneity. Investigating the upstream and downstream signaling pathways of *Cx3cr1* and their impact on epiblast segregation will be a promising area of study.

Our study newly defines the origin of CX3CR1⁺ cells. Previous reports suggested that CX3CR1⁺ cells are derived from erythro-myeloid progenitors (EMPs), which arise from the hemogenic endothelium, a mesoderm derivative (Gomez Perdiguero et al, 2015; Mass et al, 2016). However, there have been no studies that precisely trace the origin of CX3CR1⁺ cells before E8.25 utilizing *Cx3cr1* driver mice. In this study, we determined the origin of CX3CR1⁺ cells using *Cx3cr1-Cre* or *Cx3cr1-CreERT2* mice and demonstrated their presence in a subset of epiblasts at E6.5 and PE cells at E7.0. These findings provide new insight into the role of the PE in cardiogenesis. PE cells are thought to arise from the primitive endoderm that is sorted out of the inner cell mass and covers epiblasts at E3.5-E4.5 (Enders, Given et al, 1978; Gardner, 1982). At E5.5, PE cells spread out onto the inner surface of the trophectoderm and form a thick basement membrane known as Reichert's membrane (Snell and Stevens, 1966). At later stages, PE cells together with visceral endodermal cells contribute to the formation of the primary YS (Enders, Lantz et al, 1990; Enders,

Schlafke et al, 1986; Ross and Boroviak, 2020). Our findings indicate that PE cells can be derived from epiblasts as well as from the primitive endoderm, suggesting the dual origin of PE cells. Furthermore, this study offers a new insight into the developmental pathways of tissue macrophages. Previously, it was thought that EMPs give rise to tissue macrophages in the YS (Gomez Perdiguero et al, 2015), but their origin before E7.5 has been unclear. Our data demonstrate that E6.5 CX3CR1⁺ cells give rise to the PE (E7.0) that subsequently constitutes the primary YS, which produces EMPs and CX3CR1⁺ tissue macrophages at E8.5. This discovery helps to better understand the origin of tissue macrophages.

This study provides novel evidence that E6.5 CX3CR1⁺ cells are an important source of CMs that persist into adulthood. Previous reports demonstrated that the YS contains progenitors possessing cardiomyogenic potential. Murakami et al, showed that CD166⁺ cells isolated from E8.5 YS can generate CMs and ECs in vitro (Murakami et al, 2007). In addition, genetic deletion of *Scl*, a transcription factor that specifies hemogenic endothelium from mesoderm, resulted in ectopic cardiomyogenesis in the YS vasculature (Van Handel, Montel-Hagen et al, 2012). These studies suggest the existence of progenitors with cardiomyogenic potential in the early YS. However, the identity of such YS-derived progenitors and their role in heart development have been unknown. Our study revealed that E6.5 CX3CR1⁺ epiblasts differentiate into the PE that forms the YS, generating CMs as early as E9.5, when the primitive heart is already formed (Kaufman and Navaratnam, 1981). Remarkably, this late addition of cells derived from CX3CR1⁺ cells comprised about 13% of CMs in the adult heart. These data highlight the significant role of CX3CR1⁺ cells in cardiac development.

Our study further showed that CX3CR1⁺ cells represent a novel source of ECs, adding a new insight into the formation of blood vessels during cardiac development. At E5.5, angioblasts appear in the extraembryonic mesoderm (Furuta, Ema et al, 2006). At E6.5–7.5, these angioblasts aggregate to form primitive blood islands, initiating early vascular development (Drake and Fleming, 2000). At E7.5-E8.5, the nascent vascular network extends across the embryo (Drake and Fleming, 2000). At E8.5, angiogenic remodeling occurs, marked by sprouting of new vessels (Lucitti, Jones et al, 2007). Key signaling pathways, including VEGF and Notch, guide the differentiation of angioblasts into mature ECs (Hirashima, 2009). This process persists until E9.5, leading to the assembly of blood vessels within the developing heart (Vokes and Krieg, 2015). Starting at E9.5, endothelial cells undergo specialization, contributing to the formation of arteries, veins, and capillaries (Marziano, Genet et al, 2021), establishing the functional vascular network. Our data suggests that a subset of CX3CR1⁺ epiblasts at

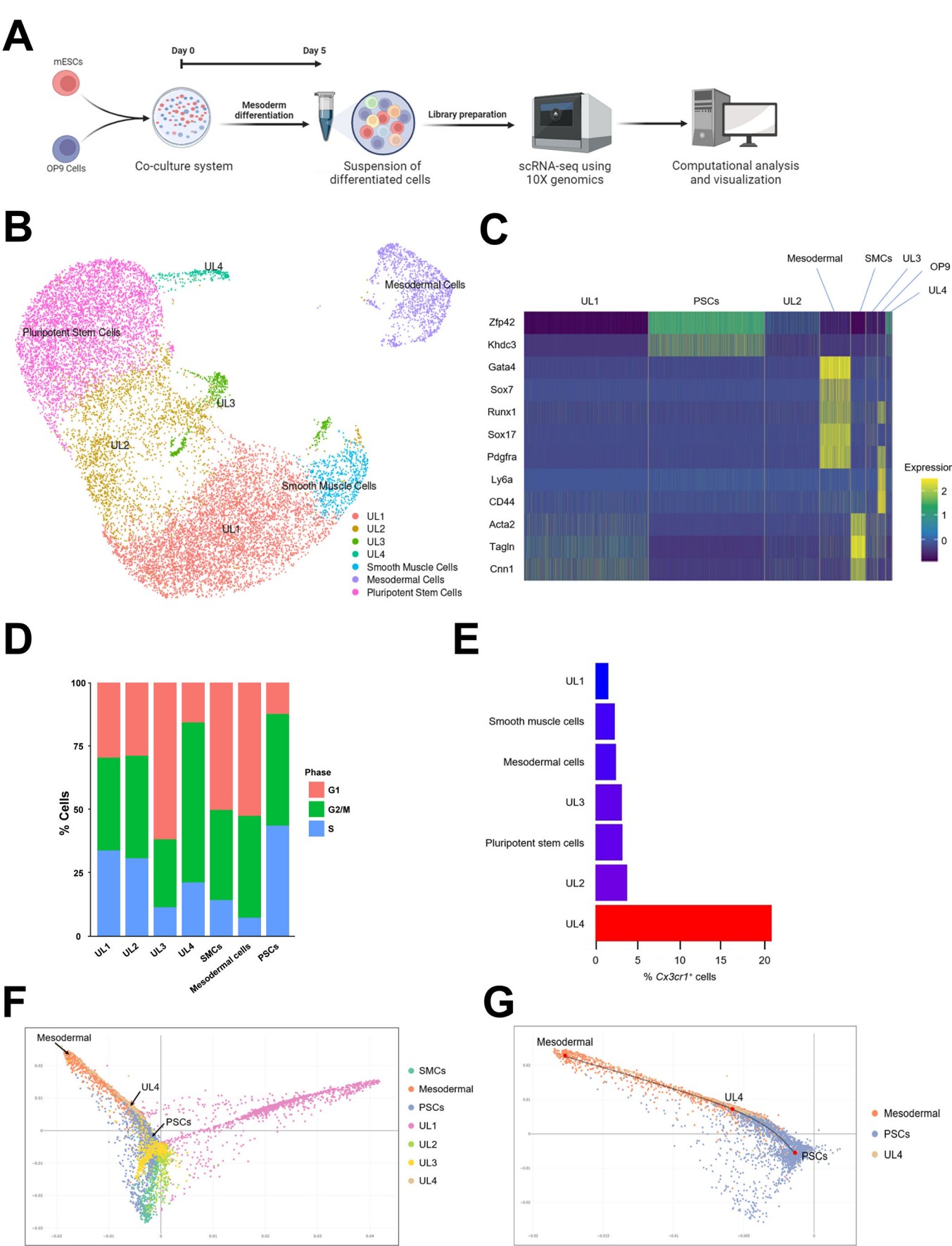

**Figure 7. scRNA-seq analysis of mESC-CX3CR1⁺ cells.**

(A) A schematic showing the experimental procedures used in this figure. J1 mESCs and OP9 cells were cocultured for 5 days and subjected to scRNA-seq. (B) UMAP visualization of 19,550 cells harvested at Day 0 (J1 mESCs) and Day 5 after coculture with OP9 cells. The cell clusters that did not express any known cell type-specific genes were defined as "unknown lineage" (UL). The cluster for OP9 cells (Ly6a⁺CD44⁺) were removed because they were not derived from mESCs. (C) A heat map showing the expression of cell type-specific genes across cell clusters. (D) Cell cycle analysis of cell clusters using the CellCycleScoring algorithm. (E) A bar graph showing the percentage of *Cx3cr1*⁺ cells in each cluster. (F) A diffusion map showing all cell clusters. (G) A diffusion map showing only mesodermal cells, UL4, and PSCs.

E6.5 is another important source of ECs. We found that CX3CR1 was not expressed in hematopoietic progenitors or endothelium in the developing heart and liver, highlighting the uniqueness of E6.5 CX3CR1⁺ cells. However, further studies are needed to determine exactly when and how ECs emerge from progeny of E6.5 CX3CR1⁺ cells. Nevertheless, it is surprising that ECs derived from CX3CR1⁺ cells and their progeny comprise ~31% of ECs in the adult heart.

This study thus suggests that CX3CR1 between E5.5 and E6.5 temporally marks multipotent progenitors capable of generating key cell types in the heart, implying its potential use for cardiovascular regeneration. CX3CR1 has been widely recognized as a marker for mononuclear phagocytes, including macrophages and monocytes (Jung, Aliberti et al, 2000). A recent study reported that CX3CR1⁺ cells in the YS are macrophage progenitors and traffic to multiple organs in the embryo, generating macrophages (Stremmel et al, 2018). However, our study demonstrates that E6.5 CX3CR1⁺ cells generate not only macrophages, but also CMs and ECs. This discovery was further validated through in vitro, ex vivo, and in vivo transplantation studies. Furthermore, our scRNA-seq analyses revealed that mESC-CX3CR1⁺ cells are an intermediate cell type that differentiates into mesodermal cells from ESCs. This multipotency of CX3CR1⁺ cells underscores their potential utility in cardiac regeneration. These findings suggest that CX3CR1⁺ cells could be a promising source for cell-based therapy and other regenerative medicine applications.

Our study suggests that CX3CR1⁺ cells undergo fusion with preexisting CMs or ECs during prenatal cardiovascular development. However, whether the fusion is transient or permanent requires further investigation using transgenic mouse models. Transient fusion allows for temporary cellular collaboration while preserving individual cell identity, thereby contributing to CM proliferation, cardiac development, and functional regeneration (Sawamiphak, Kontarakis et al, 2017). In contrast, permanent (homotypic) fusion results in the formation of multinucleated CMs (Ali, Menendez-Montes et al, 2020). Our findings suggest that the fusion observed is more likely heterotypic rather than homotypic, as tdTomato⁺ CMs were not observed when tamoxifen-pulse labeling was performed at E7.5 or E8.5—stages at which early CMs typically emerge (Brade, Pane et al, 2013).

In total, this study provides novel insight into the origin and fate of a CX3CR1⁺ multipotent progenitor population. Particularly, this study demonstrates that CX3CR1⁺ cells arise from E6.5 epiblasts, differentiate into PE cells at E7.0, and begin to contribute to cardiogenesis by generating CMs and ECs at E9.5-E10.5. The CX3CR1⁺ cells eventually make up ~13% of CMs and ~30% of ECs in the adult heart. This study further shows that ESC-derived CX3CR1⁺ cells are capable of differentiating into CMs, ECs and macrophages. This new understanding of CX3CR1⁺ cells in cardiovascular development and differentiation will help develop new strategies for disease investigation and regenerative therapy for cardiovascular disease.

## Methods

### Reagents and tools table

| Reagent/resource | Reference or source | Identifier or catalog number |
|---|---|---|
| **Experimental models** | | |
| *Cx3cr1*-cre (*M. musculus*) | The Jackson Laboratory | 025524 |
| Cx3cr1-creERT2 (*M. musculus*) | The Jackson Laboratory | 020940 |
| *R26-tdTomato* (*M. musculus*) | The Jackson Laboratory | 007909 |
| *R26-mT/mG* (*M. musculus*) | The Jackson Laboratory | 007676 |
| STO feeder cells (*M. musculus*) | ATCC | CRL-1503 |
| J1 mouse ESCs(*M. musculus*) | ATCC | SCRC-1010 |
| OP9 stromal cells (*M. musculus*) | ATCC | CRL-2749 |
| CD-1 IGS (*M. musculus*) | Charles River | 022 |
| **Antibodies** | | |
| anti-ACTN2 | Abcam | ab5694 |
| anti-MYH6 | Abcam | ab15 |
| anti-TNNT2 | Abcam | ab8295 |
| anti-CD68 | AbD Serotec | MCA19575 |
| anti-CD11b | BD | 557397 |
| anti-CD14 | BD | 553740 |
| anti-CD45 | BD | 559864 |
| anti-CDH5 | BD | 550548 |
| anti-Gr-1 | BD | 553128 |
| anti-PECAM1 | BD | 550274 |
| anti-CD41 | BioLegend | 133901 |
| anti-CX3CR1 | BioLegend | 149007 |
| anti-F4/80 | BioLegend | 123115 |
| anti-CRE | Cell Signaling Technologies | 15036 T |
| anti-KDR | Cell Signaling Technologies | 2479 |
| anti-APC magnetic beads | Miltenyi Biotec | 120-001-265 |
| anti-CD166 | Miltenyi Biotec | 130-105-444 |

| Reagent/resource | Reference or source | Identifier or catalog number |
|---|---|---|
| anti-CSF1R | Miltenyi Biotec | 130-102-962 |
| anti-Laminin | Sigma Aldrich | L9393 |
| anti-SSEA1 | BD | 560142 |
| anti-PE magnetic beads | Miltenyi Biotec | 130-048-801 |
| RNA extraction kit (RNeasy) | Qiagen | 74104 |
| Taqman reverse transcription reagents | Applied Biosystems | 4304134 |
| **Oligonucleotides and other sequence-based reagents** | | |
| Genotyping PCR primers | The Jackson Laboratory | Genotyping Protocols |
| qRT-PCR primers | Primer Bank (https://pga.mgh.harvard.edu/primerbank/) | Appendix Table S3 |
| **Chemicals, enzymes and other reagents** | | |
| Tamoxifen | Sigma Aldrich | T5648 |
| Corn oil | Sigma Aldrich | C8267 |
| BSL1 | Vector Laboratories | FL-1101 |
| CM-DiI Dye | Thermo Fisher | C7001 |
| OCT compound | Tissue-TeK | 4583 |
| Mitomycin C | Abcam | ab120797 |
| Leukemia inhibitory factor | Sigma Aldrich | L5158-5UG |
| α-MEM medium | Thermo Fisher | 12561049 |
| Accutase | Thermo Fisher | 00-4555-56 |
| BMP4 | R&D Systems | 314-BP-010 |
| VEGFA | R&D Systems | 293-VE-050 |
| CHIR99021 | Stemgent | 04-0004 |
| Insulin-transferrin-selenium | Gibco | 41400045 |
| L-ascorbic acid | Sigma Aldrich | A92902 |
| EGM2 medium | Lonza | CC-3162 |
| Matrigel | Corning | 354234 |
| DMEM/F12 medium | Thermo Fisher | 11320-033 |
| M-CSF | BioLegend | 576402 |
| DMEM high glucose medium | Thermo Fisher | 11965-092 |
| Non-essential amino acids solution | Thermo Fisher | 11140050 |
| GlutaMAX™ | Thermo Fisher | 35050061 |
| PA-RGDS | Ban et al, 2014 | N/A |
| **Software** | | |
| FlowJo | FlowJo | |
| tSNEJS | https://jefworks.github.io/tsne-online | |
| Cell Ranger | https://cloud.10xgenomics.com/cloud-analysis | |
| Seurat | https://satijalab.org/seurat | |
| ggplot2 | https://ggplot2.tidyverse.org/ | |

| Reagent/resource | Reference or source | Identifier or catalog number |
|---|---|---|
| STRING database | https://string-db.org | |
| Cytoscape | https://cytoscape.org | |
| ZEISS ZEN | ZEIZZ | |
| Applied biosystems softwares for qRT-PCR | Thermo Fisher | |
| NIS-Elements | Nikon | |
| **Other (Equipment)** | | |
| Olympus IX71 bright field microscope | Olympus | |
| Nikon Eclipse Ti fluorescence microscope | Nikon | |
| SH800S cell sorter | Sony | |
| BD LSRII flow cytometer | BD | |
| Zeiss LSM 800 Airyscan | Carl Zeiss | |

## Mice

All protocols for animal experiments were approved by the Institutional Animal Care and Use Committees (IACUC) at Emory University. All the mice used in this study were purchased from the Jackson Laboratory: *Cx3cr1*-cre (B6J.B6N (Cg)-*Cx3cr1^{tm1.1 (cre)Jung}*/J, 025524), *Cx3cr1-creERT2* (B6.129P2 (C)-*Cx3cr1^{tm2.1 (cre/ERT2)Jung}*/J, 020940), *R26-tdTomato* (Gt (ROSA)26Sor, 007909), and *R26-mT/mG* (*B6.129 (Cg)-Gt (ROSA)26Sor^{tm4 (ACTB-tdTomato,-EGFP)Luo}*/J, 007676). Genotyping was performed via standard PCR according to the instructions provided by the Jackson Laboratory. For timed matings, mice of desired genotypes were mated overnight, and vaginal plugs were checked in the following morning which was referred to as embryonic day 0.5. Embryos gained from timed mating were analyzed at the indicated time points. For pulse labeling, neonatal mice (<5 days of age) were administered 0.2 mg of tamoxifen (Sigma Aldrich, T5648) by a single subcutaneous injection. Tamoxifen was dissolved in corn oil at a concentration of 10 mg/ml. Adult mice (>8 weeks of age) were administered 125 μg per g (body weight) of tamoxifen by a single intraperitoneal injection.

## Immunohistochemistry and Immunocytochemistry

Materials including antibodies used for immunohistochemistry (IHC) and immunocytochemistry (ICC) are listed in Appendix Table S1. For IHC, mice were euthanized, and the tissues were harvested, fixed in 2% paraformaldehyde (PFA) at 4 °C overnight, and submerged in 30% sucrose solution at 4 °C overnight. Then, the tissues were mounted in OCT compound (Tissue-TeK, 4583), frozen, and cut using a Leica CM1950 cryostat. Fetal mouse hearts cultured ex vivo were harvested and treated by the same procedure. Tissue sections on glass slides were washed with PBS and permeabilized/blocked with PBS containing 0.5% Triton X-100 and 2.5% BSA at room temperature for 1 h. Samples were then incubated with primary antibodies at 4 °C overnight, washed three

times with PBS containing 0.1% Tween 20, and incubated with appropriate secondary antibodies at room temperature for 1–2 h. DAPI was used for nuclear staining. The samples were visualized under a Zeiss LSM 800 Airyscan confocal laser scanning microscope (Carl Zeiss). For visualization of blood vessels, fluorescein-labeled Griffonia (Bandeiraes) Simplicifolia Lectin I (BSL1) (Vector Laboratories, FL-1101) was injected via intracardiac route before euthanasia, as described previously (Lee, Valmiki-nathan et al, 2015). Whole mouse embryos were fixed with 2% PFA at 4 °C overnight and examined under an Olympus IX71 bright field microscope or a Nikon Eclipse Ti fluorescence microscope.

For ICC, cells were fixed in 4% PFA at room temperature for 10 min. Then, samples were washed with PBS and permeabilized/blocked with PBS containing 0.1% Triton X-100 and 2.5% BSA at room temperature for 1 h. The remainder of the procedure was the same as described above for IHC.

## Mouse ESC culture and differentiation

STO feeder cells (ATCC, CRL-1503) were treated with 10 μg/ml of Mitomycin C (Abcam, ab120797) for 2 h before culturing together with J1 mouse ESCs (ATCC, SCRC-1010). J1 mouse ESCs with STO feeder cells were maintained in high glucose DMEM containing 15% FBS and 0.2 ng/ml Leukemia inhibitory factor (LIF) on 0.1% gelatin-coated tissue culture-treated dishes. OP9 stromal cells (ATCC, CRL-2749) were maintained in α-MEM medium (Thermo Fisher, 12561049) containing 20% FBS on 0.1% gelatin-coated tissue culture-treated dishes.

To differentiate J1 mouse ESCs into CX3CR1$^+$ cells, a confluent OP9 monolayer was treated with 10 μg/ml of Mitomycin C (Abcam, ab120797) for 2 h. Then, J1 mESCs were dissociated into single cells using Accutase (Thermo Fisher, 00-4555-56), and pre-plated onto 0.1% gelatin-coated tissue culture-treated dishes for 1 h for preferential attachment of STO feeder cells. After pre-plating, non-adherent cells (mostly J1 mouse ESCs) were harvested and plated onto an OP9 monolayer at a density of $1 \times 10^5$ cells per well in 6-well plates. The coculture of J1 mESCs and OP9 was maintained in α-MEM medium containing 10% FBS, and the medium was changed every other day.

For cardiac differentiation, cells were dissociated into single cells using Accutase (Thermo Fisher, 00-4555-56) and subjected to magnetic activated cell sorting (MACS) using APC-conjugated anti-CX3CR1 (BioLegend, 149007) and anti-APC magnetic beads (Miltenyi Biotec, 120-001-265) at D5 according to the manufacturer's instructions. Sorted cells were cultured in α-MEM medium containing 20% FBS, 50 ng/ml of BMP4 (R&D Systems, 314-BP-010), 5 ng/ml of VEGFA (R&D Systems, 293-VE-050), and 4 μM of CHIR99021 (Stemgent, 04-0004). For further differentiation, cells were cultured in α-MEM medium containing 3% FBS, insulin-transferrin-selenium (Gibco, 41400045) and 50 μg/ml L-ascorbic acid (Sigma, A92902). The medium was changed every other day. For endothelial differentiation, MACS-isolated CX3CR1$^+$ cells were cultured in EGM2 medium (Lonza, CC-3162) on top of Matrigel (Corning, 354234) from D5 to D25. The medium was changed every other day. For macrophage differentiation, MACS-isolated CX3CR1$^+$ cells were cultured in DMEM/F12 medium (Thermo Fisher, 11320-033) containing 10% FBS and 20 ng/mL recombinant murine M-CSF (BioLegend, 576402) from D5 to D10. The medium was changed every other day.

## Flow cytometry

Materials including antibodies used for flow cytometry and MACS are listed in Appendix Table S2. Adult mouse CMs and ECs were isolated via the conventional Langendorff method. To quantify CMs derived from CX3CR1$^+$ cells and their progeny, isolated cardiac cells were fixed with 4% PFA for 10 min. Then, cells were permeabilized/blocked with PBS containing 0.1% Triton X-100 and 2.5% BSA at room temperature for 1 h. Next, cells were incubated with mouse anti-TNNT2 (Abcam, ab8295, 1:100) at 4 °C overnight. Cells were washed three times with PBS containing 0.1% Tween 20 and incubated with appropriate secondary antibodies at room temperature for 1–2 h. DAPI was used for nuclear staining. Cells were analyzed by a BD LSRII Flow Cytometer. The collected data were processed using FlowJo software. For EC labeling, fluorescein-labeled Griffonia (Bandeiraes) Simplicifolia Lectin I (BSL1) (Vector Laboratories, FL-1101) was injected via the intracardiac route before euthanasia and subjected to the conventional Langendorff method.

Flow cytometry for mESCs and their derivatives was conducted as previously described (Ban et al, 2014; Lee et al, 2015). Differentiating mESCs were first dissociated into single cells using Accutase (Thermo Fisher, 00-4555-56). Cells were resuspended in ice-cold PBS and incubated with fluorescently labeled antibodies. After washing twice with ice-cold PBS, cells were analyzed by a BD LSRII Flow Cytometer. The collected data were processed using FlowJo software.

## Ex vivo culture of the fetal mouse heart with mESC-derived CX3CR1$^+$ cells

E15.5 mouse hearts from CD-1 IGS pregnant mice (Charles River, 022) were cultured ex vivo as previously described (Dyer and Patterson, 2013). CX3CR1$^+$ cells were sorted from differentiating mESCs at D5 via FACS. Briefly, differentiating mESCs cocultured with OP9 cells for 5 days were dissociated into single cells using Accutase (Thermo Fisher, 00-4555-56). Cells were resuspended in ice-cold PBS and incubated with anti-CX3CR1 antibody (BioLegend, 149007) conjugated with APC at a ratio of 1:100 for 30 min on ice. After washing twice with ice-cold PBS, CX3CR1$^+$ cells were sorted using a SH800S Cell Sorter. Sorted cells were labeled with Chloromethylbenzamido (CellTracker™ CM-DiI Dye, Thermo Fisher, C7001) according to the manufacturer's instructions. DiI-labeled CX3CR1$^+$ cells were resuspended in DMEM high glucose medium (Thermo Fisher, 11965-092) containing 10% FBS, 1% non-essential amino acids solution (Thermo Fisher, 11140050), and 1% GlutaMAX™ (Thermo Fisher, 35050061). Then, cells were loaded on top of the Matrigel (Corning, 354234) at a density of $1 \times 10^5$ cells per well in a 24-well plate. DiI$^+$ cells were visualized by a Nikon Eclipse Ti fluorescence microscope. Spherical tube-like structures of DiI$^+$ cells were examined under the Olympus IX71 bright field microscope.

## Transplantation of mESC-derived CX3CR1$^+$ cells into the adult mouse heart

Cell transplantation was performed as described previously (Ban et al, 2014). We used 3-month-old male C57BL/6 mice. DiI-labeled J1 mESC-derived CX3CR1$^+$ cells ($2 \times 10^5$ cells) were encapsulated

with an injectable nanomatrix gel consisting of peptide amphiphiles and cell adhesive ligand Arg-Gly-Asp-Ser (PA-RGDS) (Ban et al, 2014; Lee et al, 2015) and were intramyocardially injected through 30 G needles at three different sites of the heart.

## Quantitative real-time PCR

J1 mESCs were dissociated into single cells using Accutase (Thermo Fisher, 00-4555-56) and subjected to MACS using PE-conjugated anti-SSEA1 (BD, 560142) and anti-PE magnetic beads (Miltenyi Biotec, 130-048-801) according to the manufacturer's instructions. For CX3CR1$^+$ cells, dissociated cells were incubated with anti-CX3CR1 antibody (BioLegend, 149007) conjugated with APC at a ratio of 1:100 for 30 min on ice. After washing twice with ice-cold PBS, cells were sorted via MACS. Then, total RNAs were isolated using a guanidinium extraction method combined with an RNA extraction kit (Qiagen, 74104). Extracted RNA was reverse-transcribed using Taqman reverse transcription reagents (Applied Biosystems, 4304134) according to the manufacturer's instructions. The synthesized cDNA was subjected to qRT-PCR using specific primers (see Appendix Table S3). Quantitative assessment of RNA levels was performed using an ABI PRISM 7500 Sequence Detection System (Applied Biosystems).

## scRNA-seq analysis

To analyze scRNA-seq data generated by Wen et al (Wen et al, 2017), we downloaded the count matrix for 236 mouse embryonic cells at E5.5–6.5 (Supplemental Table S2 in Wen et al,) and used it for generation of a t-distributed stochastic neighbor embedding (tSNE) plot with tSNEJS (https://jefworks.github.io/tsne-online). Cx3cr1$^+$ cells were defined by the expression of Cx3cr1 above mean $+1.5$ SD (standard deviation). Cx3cr1$^-$ cells were defined by the expression of Cx3cr1 below mean $-0.5$SD.

To analyze scRNA-seq data for mESC-CX3CR1$^+$ cells, raw sequencing reads (FastQ files) were processed using the 10x Genomics Cloud Analysis platform (Cell Ranger: https://cloud.10xgenomics.com/cloud-analysis). For downstream analysis, the Seurat V5 pipeline in R was utilized for demultiplexing samples, processing barcodes, quantifying single-cell gene expression, and conducting the scRNA-seq analysis. Quality control metrics included selecting cells with gene detection counts ranging from 1500 to 9000 and mitochondrial gene content below 6%. Dimensional reduction was achieved through principal component analysis, guided by a JackStraw permutation test that selected the first ten principal components with a clustering resolution set at 0.2. Differential gene expression across identified clusters was assessed using the "FindAllMarkers" function within the Seurat package. Additionally, diffusion map analysis was conducted using the Destiny package to explore data structure and relationships within the clustered single-cell data (Bian, Gong et al, 2020). The data were visualized using Seurat for uniform manifold approximation and projection (UMAP) plots and ggplot2 (https://ggplot2.tidyverse.org/). The PPI was retrieved using the STRING database (https://string-db.org), and was subsequently analyzed by Cytoscape (https://cytoscape.org).

## Statistical analysis

Initial sample size per experiment was determined based on preliminary studies, and were different for individual types of assays,

as necessary to achieve statistical significance of data. Statistical methods were not used to determine sample size. Instead, sample sizes were chosen to include at least a minimum of $(N+1)$ independent biological replicates per experiment. When comparisons between groups were made, cells and animals had been randomly assigned to the groups. Investigators were blinded to the assessment of the analyses of cell, animal, and histological experiments. To measure the fusion rate, two investigators independently performed histological analyses. All data were presented as mean ± standard error of the mean (s.e.m). For qRT-PCR data analysis, the standard unpaired Student's $t$-test (for two groups) or the one-way ANOVA test (for three or more groups) were performed.

## Data availability

The datasets and computer code produced in this study are available in the following databases: RNA-Seq data: Gene Expression Omnibus GSE271441. Modeling computer scripts: GitHub (https://github.com/kyuwoncho27/scRNAseq-for-mESC-OP9-coculture.git).

The source data of this paper are collected in the following database record: biostudies:S-SCDT-10_1038-S44318-025-00488-z.

## Peer review information

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

## Acknowledgements

We gratefully acknowledge the Emory Microscopy in Medicine (MiM) Core, the Emory Children's Pediatric Research Center flow cytometry core, and the Emory Integrated Genomics Core (EIGC). We would like to thank Dr. Ho-Wook Jun for kindly providing PA-RGDS injectable nanomatrix. This work was supported by grants from NHLBI (R01HL157242, R01HL166817, and R01HL156008). KC is a recipient of an American Heart Association postdoctoral fellowship grant (916760).

## Author contributions

**Kyuwon Cho**: Conceptualization; Formal analysis; Validation; Investigation; Visualization; Methodology; Writing—original draft. **Mark Andrade**: Formal analysis; Validation; Investigation. **Saeed Khodayari**: Data curation; Formal analysis; Investigation. **Christine Lee**: Investigation. **Seongho Bae**: Investigation. **Sangsung Kim**: Investigation; Project administration. **Jin Eyun Kim**:

Investigation. **Young-sup Yoon**: Conceptualization; Supervision; Funding acquisition; Writing—review and editing.

Source data underlying figure panels in this paper may have individual authorship assigned. Where available, figure panel/source data authorship is listed in the following database record: biostudies:S-SCDT-10_1038-S44318-025-00488-z.

## Disclosure and competing interests statement

The authors declare that they have no competing interests or relevant financial or non-financial disclosures.

