## [Peer Review File · The EMBO Journal]

Epiblast-derived CX3CR1+ progenitors generate cardiovascular cells during cardiogenesis

Kyuwon Cho, Mark Andrade, Khodayari Khodayari, Christine Lee, Seongho Bae, Sangsung Kim, Jin Kim and Young-sup Yoon

Corresponding author: Young-sup Yoon (yyoon5@emory.edu)

Review Timeline:

Submission Date:	21st Sep 24
Editorial Decision:	23rd Oct 24
Revision Received:	10th Mar 25
Editorial Decision:	14th Apr 25
Revision Received:	7th May 25
Accepted:	28th May 25

Editor: Ieva Gailite

Transaction Report:

Please note that the manuscript was previously reviewed at another journal. As EMBO Press has a transfer agreement (including the identities of the referees) with that journal, revision was invited based on the reports from that previous external submission. (Note: With the exception of the correction of typographical or spelling errors that could be a source of ambiguity, letters and reports are not edited. Depending on transfer agreements, referee reports obtained elsewhere may or may not be included in this compilation. Referee reports are anonymous unless the Referee chooses to sign their reports.)

Dear Editor,

We would like to thank the reviewers for their time and insightful comments. We were encouraged by the reviewers' positive comments. Reviewer #1 commented, "Overall, the rationale and main questions were clearly stated, and for the most part, answered. I enjoyed reading the manuscript". Reviewer #2 mentioned, "The data presented in the manuscript are intriguing and important". However, we also acknowledge that the presentation of the manuscript in places might have led to the reviewers' confusion. As mentioned below, we addressed most of the questions raised by the reviewers with further experiments and explanations. Thus, we kindly ask the editor to reconsider this manuscript for publication. Following are the point-by-point responses to the reviewers' comments.

Reviewer #1

Overall, the rationale and main questions were clearly stated, and for the most part, answered. I enjoyed reading the manuscript. Some of the most interesting data is revealed by timed tamoxifen between E6.5-E8.5 to label *Cx3cr1*⁺ cells, which leads to differential labeling and cardiomyocytes, endothelial cells and macrophages. This is both interesting, but at the same time, works against the authors own hypothesis that a multi-potent progenitor population give rise to all 3 cell types. Additional major points need to be addressed:

Response: We appreciate the reviewer's constructive comments. In general, the reviewer was positive about our studies, and we addressed the questions as follows.

1. What is the percentage of epiblast subset that express CX3CR1 at E6.5? Does this subset co-expresses NANOG (Nat Commun. 2022 Jun 21;13(1):3550.) or other epiblast pluripotency markers? (FASEB J. 2019 Jan;33(1):1179-1187; Nat Commun. 2016 Sep 2;7:12589). It is not clear whether transcriptionally and/or epigenetically CX3CR1⁺ cells represent a homogenous subset of cells with similar pluripotency, or they are different. Therefore, it is recommended to at least provide transcriptomic landscape of sorted CX3CR1⁺ at single cell level (scRNA-seq) (J Biol Chem. 2017 Jun 9;292(23):9840-9854; Nature. 2019 Dec;576(7787):487-491; Cell Stem Cell. 2021 Sep 2;28(9):1625-1640.e6). Clear identification, sorting and scRNA_seq combined with differentiation assays is required to know whether there is single multi-potential progenitor, or what is more likely the cases, different progenitors, each of which transiently express *Cx3cr1*, that are labeled simultaneously.

Response: To address the reviewer's comment, we analyzed a previously published data¹ (Wen *et al*, J Biol Chem. 2017) as recommended. In this study, embryos were collected at E5.5-6.5 and subjected to scRNA-seq. We selected this dataset because our lineage tracing data indicate that *Cx3cr1* starts to be expressed at between E5.5 and E6.5 (**Figure 1A-E**). tSNE plot clearly showed three different major cell clusters that represent epiblasts (EPI), extraembryonic ectoderm (EXE), and mesoderm (ME) (**Figure S1A-B**). Our analysis showed that 10.6% of epiblasts express *Cx3cr1* (**Figure S1A**). The tSNE plot showed that a majority of *Cx3cr1*⁺ cells (82.4%, 14 out of 17 *Cx3cr1*⁺ cells) were epiblasts. In addition, *Cx3cr1*⁺ cells highly expressed *Nanog* compared to *Cx3cr1*⁻ cells and epiblasts (**Figure S1C**). *Cx3cr1*⁺ cells also expressed other pluripotency markers such as *Pou5f1*, *Sox2*, *Dnmt3b*, *Zfp43*, *Gdf3*, and *Dppa5aat* a similar level to the epiblasts. These results are consistent with our observation that *Cx3cr1* was expressed in a subset of epiblasts at between E5.5 and E6.5. Unfortunately, Wen *et al*. removed the parietal endoderm from scRNA-seq. Thus, we were not able to perform trajectory analysis for examination of differentiation of *Cx3cr1*⁺ epiblasts to the parietal endoderm. Further immunostaining analysis showed that CX3CR1 expression turned off at E7.5 (please see comment #2 of Reviewer 2). These data suggest

that *Cx3cr1* temporally marks a subset of epiblasts that differentiate into the parietal endoderm between E5.5 and E6.5. Together, these data suggest that *Cx3cr1*⁺ cells at E6.5 consist of different progenitors with predominant epiblasts.

2. Main findings of this work rely on data generated from conditional and inducible Cre-loxP system; however

Figure S1. scRNA-seq analysis of CX3CR1⁺ cells in mouse embryos at E5.5-6.5. A-C. To examine molecular signatures of CX3CR1⁺ cells, we utilized the previously published scRNA-seq data (Wen *et al*) where embryos were collected at E5.5-6.5 and subjected to scRNA-seq. **A.** t-SNE visualization of 236 mouse embryonic cells at E5.5-6.5. Cell clusters were annotated on the basis of marker genes including *Bmp4*, *T* and *Pou5f1*. EPI, epiblast; EXE, extraembryonic ectoderm; ME, mesoderm. The numbers inside the EPI population indicate the percentage of *Cx3cr1*⁺ population out of total EPI cells (14 cells out of 132 cells). *Cx3cr1*⁺ cells were defined by the expression of *Cx3cr1* above mean + 1.5SD (standard deviation). **B.** Relative expression of *Bmp4*, *T*, and *Pou5f1* in each cluster. **C.** Relative expression of pluripotency markers in *Cx3cr1*⁺ cells, *Cx3cr1*⁻ cells and whole epiblasts. *Cx3cr1*⁺ cells were defined by the expression of *Cx3cr1* above mean + 1.5SD and *Cx3cr1*⁻ cells were defined by the expression of *Cx3cr1* below mean - 0.5SD. Error bars: standard error of mean. One-way ANOVA was performed followed by a Tukey HSD test. **P* < 0.05, ***P* < 0.01, ****P* < 0.001.

numerous pitfalls exist and need appropriate controls (Eur J Immunol. 2020 Mar;50(3):338-341; J Immunol Methods. 2014 Jun;408:89-100; Front Immunol. 2019 Sep 20;10:2228; Front Immunol. 2017 Nov 24;8:1618; Sci Rep. 2020 Sep 17;10(1):15244.). Authors need to carefully report the specificity and leakiness of their in vivo mouse models (e.g., *Cx3cr1*-*Cre* and *Cx3cr1*-*CreERT2*), and only then can appropriate conclusions be made.

Response: To examine the specificity and leakiness of *Cx3cr1-Cre* and *Cx3cr1-CreERT2* mice, we collected adult hearts and immunostained for CRE and CX3CR1 (**Figure S3A-D**). Comprehensive confocal microscopic analysis of the heart showed that expression of CRE (**Figure S3A-B**) and CREERT2 (**Figure S3C-D**) proteins was confined only to CX3CR1⁺ cells. We did not observe CX3CR1⁻ cells that express CRE or CREERT2. These results indicate that the expression of CRE and CREERT2 is specific to CX3CR1⁺ cells without noticeable leaky expression in CX3CR1⁻ cells. Thus, *Cx3cr1-Cre* and *Cx3cr1-CreERT2* mouse lines are reliable systems to specifically label CX3CR1⁺ cells.

3. The introduction focuses on the fate mapping and function of macrophages. There are called "cx3cr1 cells", as fate mapping approaches using *cx3cr1* were used to track and target them. The functional studies were done in adults. In the current work, the authors focus on developmental time points, and show using the similar fate mapping systems, cardiomyocytes and ECs are also labeled. This gives the reader the impression the prior work is confounded that the prior studies in adults targeting MFs were also target other cell types. This is not the case. The authors show here that only during early development are other cells targeted beyond Mfs are being tracked. The functional studies most studies perform in the adult using *Cx3cr1-CreERT2* would not label or target cardiomyocytes or EC. All the sections of the manuscript that deal with this, and the set up for the introduction needs to be re-written to not confuse the readers.

Response: We appreciate these constructive comments. We modified the introduction accordingly so that it does not confuse readers on pages 4-5. We specifically mentioned on page 5 that CX3CR1⁺ cells emerging from a subset of epiblasts at E6.5 prenatally contributed to generation of cardiomyocytes (CMs) and endothelial cells (ECs) in addition to macrophages.

4. As with the above statement, the authors use the generic term "cx3cr1⁺ cells" in almost every instance when referring to prior work. Much of the prior work characterize the Cx3cr1⁺ cells, and delineated there were macrophages. For example see top of page 13 "A prior study using Cx3cr1-GFP mice reported that CX3CR1⁺ cells populate the early embryonic heart as early as E9.5. However, this study was not able to identify the developmental origin, as this knock-in (Cx3cr1-GFP) mouse can only label the cells currently expressing CX3CR1 ". That study was tracking macrophages. The authors need to remove the generic phrase "cx3cr1⁺ cells" throughout their manuscript and replace it with specific information.

Response: As the reviewer suggested, we removed the part about the prior study using Cx3cr1-GFP mice and changed the generic phrase and described specific information about the cells we are referring to. When we refer to genetically labeled cells using *Cx3cr1-cre;R26-tdTomato* mice, we called them "CX3CR1⁺ cells and their progeny" (termed *Cx3cr1* lineage cells) throughout the manuscript (on pages 2, 8, 14, 15, 16, 17, 26, 36, 38, 40, and 42). When we refer to CX3CR1⁺ cells that are present at E6.5, we called them "E6.5 CX3CR1⁺ cells" (on pages 5, 17, 25, 26, and 44). When we refer to CX3CR1⁺ cells that are present during prenatal development, we called them "prenatal CX3CR1⁺ cells" (on page 16 and 19). When we refer to CX3CR1⁺ cells that are present during the postnatal period, we called them "postnatal CX3CR1⁺ cells" (on pages 15 and 16). When we refer to cells that express *Cx3cr1* transcript, we called them "*Cx3cr1*⁺ cells" (on pages 2, 5, 10, 13, 23, 48, and 51). When we refer to a general cell population that expresses *Cx3cr1* transcript or CX3CR1 protein, we called them "CX3CR1⁺ cells" throughout the manuscript.

5. Authors overstate conclusions in a number of places. "These results suggest that CX3CR1⁺ cells or their progeny contribute to cardiomyogenesis in the early developing heart as early as E9.5." The data in Fig 2 show that Cx3cr1⁺ cells are found in the developing heart. Where they contribute to cardiomyogenesis is not known.

Response: We modified the overstated conclusions accordingly. For instance, we removed "cardiomyogenesis" from places where de novo generation of CMs was not proven (on page 15). In addition, we added specific detailed information to the conclusion to avoid overgeneralization (on pages 17 and 25). Furthermore, we removed unnecessary words from the conclusion (on page 26).

6. The entire section/s of cell fusion do not seem supported by the data. When a cell that is TdTom⁺ recombines to become GFP⁺, how long does the Td signal last? Can that cell not take up (vesicle transfer) TdTomato from neighbouring cardiomyocytes or ECs? Where R26 mt/mg reporters heterozygous or homozygous? If homozygous, 1 allele could have recombined and the other not. This also does not account for leakiness of the reporter (see above). This section should be removed unless significantly strengthened.

Response: As the reviewer raised multiple concerns in this question, we would like to address them individually below.

1) Signals from remaining tdTomato protein after recombination: Fluorescent proteins generally have half-lives of 24 hrs². Thus, we speculate that the tdTomato signals would last approximately up to ~24 hrs after CRE-mediated recombination. It is not likely that all the double positive cells we observed were undergoing recombination within a 24 hr time window.

2) Vesicle transfer: As the reviewer mentioned, fluorescent proteins can be transferred to neighboring cells via extracellular vesicles (EVs). However, the fluorescence signal gained from EV uptake is minimal³. In addition, a quantitative analysis showed that EV uptake is a low yield process⁴. The spontaneous uptake rate is about 1% per hour in the beginning and this rate is reduced over time. Thus, the vesicle effect on the tdTomato signal would be negligible.

3) Zygosity: We used heterozygous *R26-mT/mG* mice, meaning that only one fluorescent reporter remains even after Cre-mediated recombination occurs.

4) Leakiness of the reporter: we examined the leakiness of the reporter mouse lines (Figure S4). We did not detect significant leaked signals in either *R26-mT/mG* or *R26-tdTomato* mice. In the *R26-mT/mG* mice, the

distance between loxP sequences is 2.4 kb, supporting that this reporter is not leaky^{5, 6}. Furthermore, this reporter mouse was previously used for assessing the cellular mechanisms of CM generation including fusion and de novo formation⁷.

7. The authors state "data not shown" in several places. In each of those places, the data needs to be shown. It is particularly relevant for Fig. 5. The authors show that E6.5 labeled cells (Td⁺) were only cardiomyocytes and ECs at E10.5, but not macrophages. The authors state they did not find macrophages at this time. This is unlikely since macrophages are already present at this time point.

Response: We included the data that were previously "not shown." Regarding Fig. 5, we stated "we did not observe tdTomato⁺ macrophages in the heart at this stage." In this experiment, tamoxifen was administered at

E6.5 to genetically label CX3CR1⁺ cells (epiblasts) with tdTomato at E6.5. We then harvested the heart at E10.5. This means that cardiac macrophages at E10.5 are not derived from CX3CR1⁺ epiblasts at E6.5. Our data do not indicate that cardiac macrophages are not present at E10.5. We also observed CD68⁺ macrophages in the E10.5 heart (**Figure 5G**).

8. The second half of the paper deals with in vitro differentiation of CX3CR1⁺ mESC cells derived from the J line. The data are interesting, however a key comparator seems to be missing. The authors either use CX3CR1⁺ cells and define them without comparing to anything and saying they can become CMs or ECs. Or they compare them to defined pluripotent cell (SSEA1⁺). When J cells are differentiating, why not remove as many pluripotent cell types and then focus on CX3CR1⁺ vs CX3CR1⁻ cells? What is special about Cx3cr1⁺ in cell culture? Can you sort this subset and show cx3cr1 expression defines a unique subpopulation? Perhaps at this stage in culture Cx3cr1⁺ cells are no different than CX3CR1⁻ cells?

Response: To characterize heterogeneity and differentiation trajectory of mESC-CX3CR1⁺ cells at the single cell level, we performed scRNA-seq (**Figure 7** and **Figure S10**). mESCs were cocultured with OP9 cells and collected at Day 5 for scRNA-seq (**Figure 7A**). Undifferentiated mESCs were also included as a control group. After filtering out low quality cells (**Figure S10A**), we performed principal component analysis (PCA) and Jackstraw analysis to identify significant principal components and to reduce dimensionality (**Figure S10B-C**). Then we utilized the uniform manifold approximation and projection (UMAP) algorithm to visualize 19,550 cells (**Figure 7B**). We identified distinct cell clusters: pluripotent stem cells, mesodermal cells, smooth muscle cells, and unknown lineages (UL) 1-4. We named cell clusters without significant expression of cell type-specific genes as “unknown lineages (UL)”. OP9 cells were removed from the UMAP. We annotated each cluster based on its expression of cell type-specific genes and this annotation was confirmed by Enrichr⁸ (**Figure 7C** and **Figure S10D**). Cell cycle analysis indicated that the UL4 showed the lowest percentage of cells at G1 phase among cell clusters differentiated from PSCs, suggesting that UL4 is the most proliferative cell cluster after PSCs (**Figure 7D**). To identify cell clusters that express *Cx3cr1*, we calculated the percentage of *Cx3cr1*⁺ cells, and the UL4 showed the highest percentage (**Figure 7E**). To define differentiation trajectories, we performed diffusion map analysis⁹. Three tips and three branches were detected, suggesting that mESCs were differentiated into three major different lineages (**Figure S10E**). Further analysis showed that the UL4 was aligned in between pluripotent stem cells and mesodermal cells (**Figure 7F-G**). These results demonstrate that *Cx3cr1*⁺ cells represent a unique intermediate cell population differentiating to the mesoderm from pluripotent stem cells (PSCs).

Figure 7. scRNA-seq analysis of mESC-CX3CR1⁺ cells. **A.** A schematic showing the experimental procedures used in this figure. J1 mESCs and OP9 cells were cocultured for 5 days and subjected to scRNA-seq. **B.** UMAP visualization of 19,550 cells harvested at Day 0 (J1 mESCs) and Day 5 after coculture with OP9 cells. Clusters that do not express any known cell type-specific genes were defined as an 'unknown lineage' (UL). The cluster for OP9 cells (Ly6a⁺CD44⁺) were removed because they were not derived from mESCs. **C.** A heat map showing the expression of cell type-specific genes across cell clusters. **D.** Cell cycle analysis of cell clusters using the CellCycleScoring algorithm. **E.** A bar graph showing the percentage of *Cx3cr1*⁺ cells in each cluster. **F.** A diffusion map showing all cell clusters. **G.** A diffusion map showing only mesodermal cells, UL4, and PSCs.

Figure S10. scRNA-seq analysis of differentiating mESCs cultured with OP9 cells at D5. J1 mESCs and OP9 cells were cocultured for 5 days and subjected to scRNA-seq as shown in Figure 7A. **A.** Filtering out low-quality cells. Cells with total number of feature RNA between 1,500 and 9,000 were included for further analysis (left). In addition, cells with more than 6% of mitochondrial genes were excluded for further analysis (right). **B.** Principal component analysis (PCA). **C.** Jackstraw permutation test. **D.** A heatmap for differentially expressed genes (DEGs) across all clusters. The top 10 differentially expressed genes (DEGs) were identified using a significance threshold set at \log_2FC (Fold Change) ≥ 0.25 and P value ≤ 0.05 . **E.** A diffusion map showing three major differentiation trajectories.

9. The singular use of CSF1R only to exclude myeloid potential, and CD166 to state CM and EC potential is not sufficient. How do you know the antibody worked, it is notoriously difficult to see by flow cytometry, especially if tissue is digested. Additional markers need to be used (Fig S8). Gene expression studies are interesting, but additional genes are needed, in particular myeloid genes.

Response: For validation of the antibodies, we utilized isotype controls as negative controls and performed flow cytometry with *in vitro* cultured cells, not tissue-digested cells. Since CX3CR1 is known as a marker for myeloid cells¹⁰, we checked whether mESC-CX3CR1⁺ cells express myeloid markers such as CD11b, CD14, F4/80, and Gr-1 via flow cytometry. The results showed that mESC-CX3CR1⁺ cells did not express these markers (**Figure S6C**). We also checked the expression of genes related to myeloid differentiation such as *Tal1* (a transcription factor required for differentiation of mesoderm to hematopoietic stem cells), *Cebpa* (a transcription factor required for differentiation of hematopoietic stem cells to granulocyte-monocyte progenitors), and *Irf8* (a transcription factor required for differentiation of monoblasts to macrophages)¹¹ via quantitative reverse transcription polymerase chain reaction (qRT-PCR). We used J1 mouse ESCs as a negative control and adult mouse bone marrow (BM) as a positive control. We found that mESC-CX3CR1⁺ cells minimally express these markers without significant difference from J1 mESCs (**Figure S6D**). These results suggest that mESC-CX3CR1⁺ cells are not myeloid lineage cells.

10. Image Reproduction - the same image has been used in main (Figure 2C-E) and supplementary (Figure S2A-E) figures. Therefore, it is recommended to provide all images for each experiment.

Response: It appears that the reviewer might have misunderstood our intention. We intentionally used Figure 2A and Figure S2A with the same image to facilitate the readers' understanding. In this way, we can show both the entire area and the specific areas where we have selected the images from Figure 2F and Figure S2C-E. Nevertheless, we removed E9.5 heart figures from the supplementary documents to avoid any potential misunderstanding.

11. In Figure 3D, were immune cells (CD45⁺ cells) excluded? as the Td⁺TNNT2⁺ cells may also represent macrophages engulfing CMs?

Response: We immunostained the tissue for CD45 (Figure S2F) and excluded them from the calculation of the fusion rate. The result was the same as before.

12. Is there any reason why authors choose ACTN2 in some images while in others TNNT2 was used? Make sure the text and image labeling align to each other, e.g., Figure 3E.

Response: Both markers are widely and interchangeably used for cardiomyocyte staining. Regarding Figure 3E, we corrected the labeling accordingly.

13. It is suggested to reduce and re-write the result section from p16 “Mouse ESC-derived CX3CR1⁺ cells generate CMs and ECs in vitro”. There are too many details on the employed protocol unrelated to the results.

Response: We modified and reduced the Results section accordingly. We removed unnecessary details accordingly on page 20.

Reviewer #2 (Remarks to the Author):

In this manuscript, Cho et al. study the contribution of CX3CR1⁺ cells to the different cardiac cell types during embryonic development, postnatal heart growth, and adult homeostasis using constitutive and inducible *Cx3cr1-Cre* driver lines. Tissue-resident macrophages (derived from embryonic CX3CR1⁺ precursors) have recently gained attention due to their important function in tissue healing and regeneration, in contrast to macrophages originating from definitive hematopoietic progenitors that play mainly role in inflammatory processes. In this study, Cho et al show that CX3CR1⁺ precursors arise in the epiblast as early as E6.5 and contribute not only to resident macrophages but also to cardiomyocytes and endothelial cells during embryonic development. Importantly the authors show that ESC-derived CX3CR1⁺ precursors are capable of differentiating in macrophages, cardiomyocytes, and endothelial cells in cell culture as well as populating ex vivo cultured hearts and differentiating in these distinct cardiac cell types. While the data presented in the manuscript are intriguing and important, additional experiments are needed to clarify the developmental trajectories of CX3CR1⁺ precursors and their progeny.

1) The manuscript is mainly based on lineage tracing of CX3CR1⁺ cells. While lineage tracing studies have been instrumental in gaining critical insights into cellular origins and developmental plasticity, single-cell technologies have reshaped the area of developmental biology and provided new critical insights into the developmental trajectories and cell fates during embryogenesis. Since CX3CR1⁺ cells are found in the epiblast, the parietal endoderm, and the yolk sac it would be important to show how these cell types are related and give rise to cardiomyocytes and endothelial cells. There is a growing number of resource manuscripts providing the transcriptome landscape of single cells during embryogenesis (Pijuan-Sala B, et al, Nature, 566:490-5, 2019; Cao J, et al., Nature, 566:496-502, 2019.), which could be used to further clarify the developmental trajectories of CX3CR1⁺ cells originating in the epiblast, the yolk sac and the endoderm.

Response: As both Reviewer 1 and 2 suggested, we utilized one previously published scRNA-seq data to examine embryonic CX3CR1⁺ cells. Please refer to comment #1 of Reviewer 1.

2) Is *Cx3cr1* expressed in the liver bud or the hemogenic angioblasts (Nakano, H. et al., Nat. Commun. 4, 1564, 2013 and Zamir et al., Elife, 2017 Mar 8;6:e20994.), which give rise to the endocardium? The endocardium has remarkable plasticity, capable of generating cardiomyocytes, coronary endothelial cells as well as blood cells. More careful characterization of *Cx3cr1* expression (e.g. in situ hybridization at different embryonic stages) will be instrumental to interpret the results from the lineage tracing analysis.

Response: To examine expression of CX3CR1 in the developing endocardium, we performed immunostaining with an anti-CX3CR1 antibody. We used PECAM1 as an endothelium marker and CD41 as an early hematopoietic progenitor marker^{12, 13}. In the developing heart, CX3CR1 was not colocalized with PECAM1⁺ endocardial cells or CD41⁺ hemogenic cells (**Figure S5A**). As a positive control, we stained the developing heart for CD68 and detected its colocalization with CX3CR1 at the epicardium area. Liver bud is another embryonic organ for active hematopoiesis¹⁴. Thus, we examined whether CX3CR1 is expressed in the endothelium or early hematopoietic progenitors in the developing liver (**Figure S5B**). Confocal microscopic analyses showed that CX3CR1 was not detected on PECAM1⁺ endothelial cells or CD41⁺ hemogenic cells in the liver bud. As a positive control, we stained the developing liver for CD68 and detected its colocalization with CX3CR1. Together, these results suggest that CX3CR1⁺ cells are not derived from the hemogenic populations in the heart or the liver bud. Furthermore, we checked the expression pattern of CX3CR1 at different embryonic stages using an anti-CX3CR1 antibody. We found that CX3CR1 was transiently expressed at E6.5 and turned off at E7.5 (**Figure S5C**). At E8.5-9.5, we observed CX3CR1⁺ cells in the developing embryo, which is consistent with previous reports^{15, 16}. These data together with our genetic lineage tracing data (**Figure 1**) suggest that CX3CR1 temporally marks a subset of epiblasts at between E5.5 and E6.5.

3) It would be interesting to analyze the contribution of the different CX3CR1⁺ precursors originating in the epiblast, the yolk sac, and the endoderm (sorted at different time points) to the different cardiac lineages using cell culture-based systems/ex vivo embryonic heart colonization approaches.

Response: Thank you for suggesting an interesting study. However, there are practical challenges that limit the feasibility of this approach. First, our data show that CX3CR1 is transiently expressed on a subset of epiblasts at E6.5 (**Figure 1E** and **Figure S5C**, please see comment #2 of Reviewer 2). We found that CX3CR1 is not expressed on the parietal endoderm at E7.5 or the yolk sac at E8.5, making it impractical to sort CX3CR1⁺ cells at these specific time points. Second, conventional sorting methods including FACS and MACS do not guarantee a 100% pure CX3CR1⁺ cell population. The possibility of including a small fraction of CX3CR1-negative progenitor cells in the sorted population can confound the interpretability of the data. The potential presence of these negative cells could introduce ambiguity in the observed cardiac lineage differentiation. Finally, our culture-based systems and *ex vivo* embryonic heart colonization approaches have inherent limitations when compared to the *in vivo* microenvironment. As demonstrated in Figure 1, genetic lineage tracing *in vivo* has confirmed the ability of CX3CR1⁺ cells to differentiate into cardiac lineages. Given the reliability of this *in vivo* system, we do not think that pursuing similar investigations with our *in vitro* or *ex vivo* systems would not provide any additional benefits.

4) More careful characterization of the ESC-derived CX3CR1⁺ cell population is required. Do these cells express *Isl1* and *Nkx2-5* and to what extent? Are there different CX3CR1⁺ cell populations during the course of ESC differentiation? The qPCR results indicate that there are indeed different cell populations, thus it would be important to clarify the molecular signature of the CX3CR1⁺ cell population that can give rise to macrophages, cardiomyocytes, and endothelial cells. Here single cell RNA Seq particularly at day 5, which was used to sort for multipotent macrophage, cardiomyocyte, and endothelial cell precursors, would be helpful.

Response: Since our genetic lineage tracing data suggest that CX3CR1⁺ cells contribute to CMs and ECs (**Figure 5**), we checked whether mESC-CX3CR1⁺ cells express markers for cardiac progenitors such as *Nkx2-5*, *Gata4*, *Isl1*, *Tbx5* and *Mef2c*. We used J1 mouse ESCs as a negative control and E10.5 mouse hearts as a positive control. We found that mESC-CX3CR1⁺ cells showed higher expression of *Nkx2-5* and *Gata4* compared to J1 mESCs and CX3CR1⁻ cells (**Figure S6E**). For *Isl1*, *Tbx5* and *Mef2c*, there was no significant difference between CX3CR1⁺ cells and CX3CR1⁻ cells. These data suggest that mESC-CX3CR1⁺ cells have partial characteristics of cardiac progenitors and are more similar to the first heart progenitor (*Nkx2-5*⁺) than the second heart progenitor (*Isl1*⁺). To further clarify the molecular signature of CX3CR1⁺ cells, we performed single cell RNA-seq for mESC-derived CX3CR1⁺ cells. Please refer to comment #8 of Reviewer 1.

Figure S6E. Expression of cardiac transcription factors in differentiating mESCs cultured with OP9 cells at D5. Gene expression profile of cardiac transcription factors in differentiating mESCs cultured with OP9 cells at D5. Differentiating mESCs at Day 5 were subjected to MACS using an anti-CX3CR1 antibody. Sorted cells (CX3CR1⁺ cells and CX3CR1⁻ cells) together with J1 mESCs and E10.5 mouse hearts were subjected to RNA extraction and qRT-PCR. Error bars: standard error of mean. One-way ANOVA was performed followed by a Tukey HSD test. *P < 0.05, **P < 0.01, ***P < 0.001. BM, bone marrow.

5) Regarding the contribution of CX3CR1⁺ to cardiomyocytes and endothelial cells through cell fusion: Although using R26-mT/mG reporter in combination with Cre drivers is a valid approach, I am wondering whether allele variations (as reported for *Cx3cr1*) could result in recombination of only one allele and thereby double positivity for GFP and tdTomato? Does cell fusion contribute to the age-dependent decline of CX3CR1⁺ resident macrophages, potentially hindering cardiac regeneration? Some discussion of these roles would be helpful.

Response: We crossed the *Cx3cr1-Cre* mouse to the mT/mG mouse. Thus, the resultant mice are necessarily heterozygous for mT/mG. This means that these mice have the mT/mG reporter gene on only one allele. In the reporter-containing allele, the Cre recombinase cuts out the loxP-mT-stop-loxP sequence, leading to expression of mG. However, the other allele does not express any fluorescent protein because it does not contain any reporter gene. These mechanisms prevent dual expression of mT and mG at the same time. A report showed that the CX3CR1⁺ macrophage population progressively declines in the heart as the mice get older¹⁷. We did not observe fusion between CX3CR1⁺ cells and cardiomyocytes when we pulse-labeled CX3CR1⁺ cells at 3 months of age. But it is still possible that CX3CR1⁺ macrophages could fuse with other cells at an earlier stage (before 3 months old). If that were the case, the fusogenic characteristics of CX3CR1⁺ macrophages might account for the age-dependent decline of CX3CR1⁺ resident macrophages and limited regenerative capacity of the adult heart.

Reference

1. Wen J, Zeng Y, Fang Z, Gu J, Ge L, Tang F, Qu Z, Hu J, Cui Y, Zhang K, Wang J, Li S, Sun Y and Jin Y. Single-cell analysis reveals lineage segregation in early post-implantation mouse embryos. *J Biol Chem*. 2017;292:9840-9854.
2. Snapp EL. Fluorescent proteins: a cell biologist's user guide. *Trends Cell Biol*. 2009;19:649-55.
3. Russell TB, Skinner AM and Kurre P. Programmed vesicle transfer of green fluorescent protein from a stably transduced cell line to primary hematopoietic cells. *Blood*. 2012;119:5330-2.
4. Bonsergent E, Grisard E, Buchrieser J, Schwartz O, They C and Lavieu G. Quantitative characterization of extracellular vesicle uptake and content delivery within mammalian cells. *Nat Commun*. 2021;12:1864.
5. Stifter SA and Greter M. STOP floxing around: Specificity and leakiness of inducible Cre/loxP systems. *Eur J Immunol*. 2020;50:338-341.
6. Alvarez-Aznar A, Martinez-Corral I, Daubel N, Betsholtz C, Makinen T and Gaengel K. Tamoxifen-independent recombination of reporter genes limits lineage tracing and mosaic analysis using CreER(T2) lines. *Transgenic Res*. 2020;29:53-68.
7. van Berlo JH, Kanisicak O, Maillet M, Vagnozzi RJ, Karch J, Lin SC, Middleton RC, Marban E and Molkentin JD. c-kit⁺ cells minimally contribute cardiomyocytes to the heart. *Nature*. 2014;509:337-41.
8. Chen EY, Tan CM, Kou Y, Duan Q, Wang Z, Meirelles GV, Clark NR and Ma'ayan A. Enrichr: interactive and collaborative HTML5 gene list enrichment analysis tool. *BMC Bioinformatics*. 2013;14:128.
9. Haghverdi L, Buettner F and Theis FJ. Diffusion maps for high-dimensional single-cell analysis of differentiation data. *Bioinformatics*. 2015;31:2989-98.
10. Gordon S and Taylor PR. Monocyte and macrophage heterogeneity. *Nat Rev Immunol*. 2005;5:953-64.
11. Rosenbauer F and Tenen DG. Transcription factors in myeloid development: balancing differentiation with transformation. *Nat Rev Immunol*. 2007;7:105-17.
12. Mitjavila-Garcia MT, Cailleret M, Godin I, Nogueira MM, Cohen-Solal K, Schiavon V, Lecluse Y, Le Pesteur F, Lagrue AH and Vainchenker W. Expression of CD41 on hematopoietic progenitors derived from embryonic hematopoietic cells. *Development*. 2002;129:2003-13.
13. Zovein AC, Hofmann JJ, Lynch M, French WJ, Turlo KA, Yang Y, Becker MS, Zanetta L, Dejana E, Gasson JC, Tallquist MD and Iruela-Arispe ML. Fate tracing reveals the endothelial origin of hematopoietic stem cells. *Cell Stem Cell*. 2008;3:625-36.
14. Gordillo M, Evans T and Gouon-Evans V. Orchestrating liver development. *Development*. 2015;142:2094-108.
15. Schulz C, Gomez Perdiguero E, Chorro L, Szabo-Rogers H, Cagnard N, Kierdorf K, Prinz M, Wu B, Jacobsen SE, Pollard JW, Frampton J, Liu KJ and Geissmann F. A lineage of myeloid cells independent of Myb and hematopoietic stem cells. *Science*. 2012;336:86-90.
16. Epelman S, Lavine KJ, Beaudin AE, Sojka DK, Carrero JA, Calderon B, Brija T, Gautier EL, Ivanov S, Satpathy AT, Schilling JD, Schwendener R, Sergin I, Razani B, Forsberg EC, Yokoyama WM, Unanue ER, Colonna M, Randolph GJ and Mann DL. Embryonic and adult-derived resident cardiac macrophages are maintained through distinct mechanisms at steady state and during inflammation. *Immunity*. 2014;40:91-104.
17. Molawi K, Wolf Y, Kandalla PK, Favret J, Hagemeyer N, Frenzel K, Pinto AR, Klapproth K, Henri S, Malissen B, Rodewald HR, Rosenthal NA, Bajenoff M, Prinz M, Jung S and Sieweke MH. Progressive replacement of embryo-derived cardiac macrophages with age. *J Exp Med*. 2014;211:2151-8.

Dear Dr. Yoon,

Thank you for submitting your manuscript together with the reviews from another journal and your point-by-point response to them to The EMBO Journal. I have now received input from both original reviewers. I have copied their comments below.

As you can see, both reviewers appreciate that the improvements in the revised version. However, they also find that the statements regarding cell fusion need to be toned down in the absence of more conclusive data. Furthermore, reviewer #2 finds that the data regarding potential tamoxifen-independent leakiness of the CreERT2 mouse line need to be demonstrated before he/she can recommend acceptance of the study.

Based on this input, I would therefore invite you to address these remaining comments in a revised manuscript. In particular, the control experiments requested by reviewer #2 will be crucial for acceptance here. I would also be happy to discuss the revision in more detail via email or phone/videoconferencing.

We generally allow three months as standard revision time, which can be extended up to six months for major revisions. Should you foresee a problem in meeting this deadline, please let us know in advance to discuss an extension.

As a matter of policy, competing manuscripts published during this period will not negatively impact on our assessment of the conceptual advance presented by your study. However, please contact me as soon as possible upon publication of any related work to discuss the appropriate course of action.

When preparing your letter of response to the referees' comments, please bear in mind that this will form part of the Review Process File and will therefore be available online to the community. For more details on our Transparent Editorial Process, please visit our website: <https://www.embopress.org/page/journal/14602075/authorguide#transparentprocess>. Please also see the attached instructions for further guidelines on preparation of the revised manuscript.

Please feel free to contact me if have any further questions regarding the revision. Thank you for the opportunity to consider your work for publication.

With best regards,

leva

leva Gailite, PhD
Senior Scientific Editor
The EMBO Journal
Meyerhofstrasse 1
D-69117 Heidelberg
Tel: +4962218891309
i.gailite@embojournal.org

- a point-by-point response to the referees' comments, with a detailed description of the changes made (as a word file).
- a word file of the manuscript text.

- individual production quality figure files (one file per figure)
 - a complete author checklist, which you can download from our author guidelines (<https://www.embopress.org/page/journal/14602075/authorguide>).
 - Expanded View files (replacing Supplementary Information)
- Please see out instructions to authors
<https://www.embopress.org/page/journal/14602075/authorguide#expandedview>
- a Reagents and Tools Table as part of the Methods section, which can be downloaded from our author guidelines (<https://www.embopress.org/page/journal/14602075/authorguide#structuredmethods>)

We realize that it is difficult to revise to a specific deadline. In the interest of protecting the conceptual advance provided by the work, we recommend a revision within 3 months (21st Jan 2025). Please discuss the revision progress ahead of this time with the editor if you require more time to complete the revisions.

Referee #1:

The revised manuscript has addressed most of my comments and is improved. Significant new data have been added that enhance the conclusions. Overall, the work convincingly shows that CX3CR1 epiblast cells contribute to cardiomyocytes and endothelial cells in the heart. However, I am still not entirely convinced by the results on cell fusion, which show that around 20% of CX3CR1+ cells-derived cardiomyocytes and endothelial cells fuse with CX3CR1+ cells. Is this fusion transient, or are the tdTomato+GFP+ double-positive cells multinucleated? Is this homotypic or heterotypic fusion? For example, homotypic fusion of cardiomyocytes results in multinucleated cardiomyocytes after birth (PMID: 32510998). Additionally, the different populations identified in the single-cell sequencing experiments presented in Figure 7 would benefit from more extensive analysis (two large populations are referred to as unknown lineage). CX3CR1 is enriched in unknown lineage 4; so it would be important to analyze the transcriptional signatures and identify the in vivo equivalent of this population more thoroughly. The gastrulation model in PMID: 33932341 would be helpful for the analysis. Lastly, the sentence in the abstract needs to be revised or removed: "Here, we performed genetic lineage tracing of CX3CR1+ cells and their progeny (termed Cx3cr1 lineage cells) and demonstrated that they emerge from a subset of epiblasts at E6.5 and differentiate into the parietal endoderm at E7.0." Written this way, it gives the impression that the parietal endoderm is derived from the epiblast, which is not the case. This aspect has generally not been characterized thoroughly.

Referee #2:

The authors have submitted a revised manuscript I have previously reviewed. The authors explore the use of the Cx3cr1 fate mapping system to define the progeny of cells that express Cx3cr1.

Fusion

- In the description of in the results of Fig 3E in the results, the authors refer to the mouse as CreERT2, but it appears to be a Cre only. The fact there are cells that are double labelled does not indicate fusion - that statement needs to be removed. As brought up in the prior review, it could be a number of reasons, no of which were assessed.

Leakiness

- this implies tamoxifen independent expression of reporter of the CreERT2 mice. The authors need to do these studies to show that in Cx3cr1-CreERT2, what cells are labeled when no tamoxifen is given. It is well appreciated that the Cx3cr1-CreERT2 are leaky if given enough time. The authors data in this regard is sufficient - we need to see Cx3cr1-CreERT2 : R26 Td OR Cx3cr1-CreERT2 : R26 mt/mg mice without tamoxifen at the time points used in the study when the experiments above are performed

The in Fig S9 is not well explained or explored. There are several parts where ev vivo cultured embryonic hearts are used. This

needs better explanation and quantitation of the results, rather and representative images. Find a single cell that remains after culture is not sufficient.

Overall, the paper is an improvement. The key experiments that are needed are those where the Cx3cr1CrERT2 is given tamoxifen at a few times in development and we see if cardiomyocytes and endothelial cells are labelled in adult mice - this was not done as far as I can see, yet this is essential. In those experiments, mice that did not receive tamoxifen need to be used as controls to assess for leakiness.

Re: Regarding the manuscript entitled “**CX3CR1⁺ cells originate from epiblasts and generate cardiovascular cells during cardiogenesis**”

Dear Editor,

We would like to thank the reviewers for their time and insightful comments. We have done our best to adequately address the questions raised by the reviewers. Thus, we kindly ask the editor to consider this manuscript for publication. Below are our point-by-point responses to the reviewers' comments. The modification is highlighted in blue in the revised manuscript.

Referee #1:

1. The revised manuscript has addressed most of my comments and is improved. Significant new data have been added that enhance the conclusions. Overall, the work convincingly shows that CX3CR1 epiblast cells contribute to cardiomyocytes and endothelial cells in the heart. However, I am still not entirely convinced by the results on cell fusion, which show that around 20% of CX3CR1+cells-derived cardiomyocytes and endothelial cells fuse with CX3CR1+ cells. Is this fusion transient, or are the tdTomato+GFP+ double-positive cells multinucleated? Is this homotypic or heterotypic fusion? For example, homotypic fusion of cardiomyocytes results in multinucleated cardiomyocytes after birth (PMID: 32510998).

Response: Our study suggests that CX3CR1⁺ cells undergo fusion with pre-existing CMs or ECs during prenatal cardiovascular development. However, whether the fusion is transient or permanent requires further investigation using transgenic mouse models. Transient fusion allows for temporary cellular collaboration while preserving individual cell identity, thereby contributing to CM proliferation, cardiac development, and functional regeneration¹. In contrast, permanent (homotypic) fusion results in the formation of multinucleated CMs². Our findings suggest that the fusion observed is more likely heterotypic rather than homotypic, as tdTomato⁺ CMs were not observed when tamoxifen-pulse labeling was performed at E7.5 or E8.5—stages at which early CMs typically emerge³. We included this in the Discussion on page 28.

2. The different populations identified in the single-cell sequencing experiments presented in Figure 7 would benefit from more extensive analysis (two large populations are referred to as unknown lineage). CX3CR1 is enriched in unknown lineage 4; so it would be important to analyze the transcriptional signatures and identify the *in vivo* equivalent of this population more thoroughly. The gastrulation model in PMID: 33932341 would be helpful for the analysis.

Response: To analyze transcriptional signatures of *Cx3cr1*⁺ cells, we selected *Cx3cr1*⁺ cells from UL4 cluster and examined their protein-protein interaction (PPI) network using STRING database and was subsequently analyzed by Cytoscape program (Appendix Figure S10F). We found that the core network of *Cx3cr1*⁺ cells comprises transcription factors including *Zfp352*, *Zscan4*, and *Tcstv* (Appendix Figure S10G and Appendix Figure S10H). *Zfp352* activates developmental programs during embryogenesis⁴. *Zscan4* has multiple functions including derepression of heterochromatin, maintenance of telomere length, and genome stability in mESCs^{5,6}. *Tcstv* elongates telomeres of mESCs⁷. These reports suggest that *Cx3cr1*⁺ cells have features of both differentiating cells and mESCs. This supports our notion that *Cx3cr1*⁺ cells represent a transitional differentiating cell population exiting pluripotent state, and their *in vivo*

equivalent would be a subset of primed epiblasts emerging at between E5.5 and E6.5. The data was added to Appendix Figure S10 and the explanation was added to page 24.

3. The sentence in the abstract needs to be revised or removed: "Here, we performed genetic lineage tracing of CX3CR1+ cells and their progeny (termed *Cx3cr1* lineage cells) and demonstrated that they emerge from a subset of epiblasts at E6.5 and differentiate into the parietal endoderm at E7.0." Written this way, it gives the impression that the parietal endoderm is derived from the epiblast, which is not the case. This aspect has generally not been characterized thoroughly.

Response: Our data demonstrate that a subset of parietal endoderm (PE) is derived from the epiblast (Figure 1). *TdTomato*⁺ cells, labeled in a subset of epiblasts at E6.5, are exclusively found in the PE at E7.5 and not in the epiblast. Given that this is a permanent genetic labeling system, it is unlikely that *tdTomato*⁺ cells would vanish from the E7.5 epiblast after being labeled at E6.5.

Referee #2:

The authors have submitted a revised manuscript I have previously reviewed. The authors explore the use of the *Cx3cr1* fate mapping system to define the progeny of cells that express *Cx3cr1*.

1. Fusion- In the description of in the results of Fig 3E in the results, the authors refer to the mouse as CreERT2, but it appears to be a Cre only. The fact there are cells that are double

labelled does not indicate fusion - that statement needs to be removed. As brought up in the prior review, it could be a number of reasons, no of which were assessed.

Response: The label “CreERT2” on page 15 is a typographical error and was corrected. The Molkentin group’s Nature paper on genetic lineage tracing of c-kit⁺ cells used the same system and reached the same conclusions (van Berlo et al, Nature, 2014). Notably, we benchmarked their study to design our experiments for the fusion analysis.

2. Leakiness- this implies tamoxifen independent expression of reporter of the CreERT2 mice. The authors need to do these studies to show that in Cx3cr1-CreERT2, what cells are labeled when no tamoxifen is given. It is well appreciated that the Cx3cr1-CreERT2 are leaky if given enough time. The authors data in this regard is sufficient - we need to see Cx3cr1-CreERT2:R26 Td OR Cx3cr1-CreERT2:R26 mt/mg mice without tamoxifen at the time points used in the study when the experiments above are performed

Response: As shown in Figures 3D and 4D, tamoxifen treatment does not result in labeling of CMs or ECs. Therefore, it is reasonable to infer that CM or EC labeling would not occur under conditions without tamoxifen.

Since CX3CR1 is strongly expressed in macrophages, there may be weak leaky signals observed in the macrophage population. However, this would not impact our conclusion that CX3CR1⁺ cells contribute to CMs and ECs. In addition, previous studies using similar systems have well-established that CX3CR1⁺ cells (post E8.5) contribute to cardiac macrophages, which aligns with our findings (Figure 5G and Figure 5J)^{8,9}.

3. The in Fig S9 is not well explained or explored. There are several parts where ex vivo cultured embryonic hearts are used. This needs better explanation and quantitation of the results, rather than representative images. Finding a single cell that remains after culture is not sufficient.

Response: The contribution of CX3CR1⁺ cells to macrophages is well-documented, and this confirmatory experiment was conducted to replicate previously established findings. Representative data were presented for this purpose. To address the reviewer’s concerns, we included low-magnification images in Appendix Figure S9.

4. Overall, the paper is an improvement. The key experiments that are needed are those where

the Cx3cr1CreERT2 is given tamoxifen at a few times in development and we see if cardiomyocytes and endothelial cells are labelled in adult mice - this was not done as far as I can see, yet this is essential. In those experiments, mice that did not receive tamoxifen need to be used as controls to assess for leakiness.

Response: Administering tamoxifen during development results in abortion, making the proposed experiments unfeasible. In our hands, even with progesterone co-treatment, abortion typically occurred within five days. Thus, tamoxifen treatment was limited to short-term experiments: we sacrificed mice within five days of administration. Administering tamoxifen multiple times would make the experiment even more impractical.

Reference

1. Sawamiphak, S., Kontarakis, Z., Filosa, A., Reischauer, S. & Stainier, D.Y.R. Transient cardiomyocyte fusion regulates cardiac development in zebrafish. *Nat Commun* **8**, 1525 (2017).
2. Ali, S.R., Menendez-Montes, I., Warshaw, J., Xiao, F. & Sadek, H.A. Homotypic Fusion Generates Multinucleated Cardiomyocytes in the Murine Heart. *Circulation* **141**, 1940-1942 (2020).
3. Brade, T., Pane, L.S., Moretti, A., Chien, K.R. & Laugwitz, K.L. Embryonic heart progenitors and cardiogenesis. *Cold Spring Harb Perspect Med* **3**, a013847 (2013).
4. Mwalilino, L., *et al.* The role of Zfp352 in the regulation of transient expression of 2-cell specific genes in mouse embryonic stem cells. *Genes Cells* **28**, 831-844 (2023).
5. Akiyama, T., *et al.* Transient bursts of Zscan4 expression are accompanied by the rapid derepression of heterochromatin in mouse embryonic stem cells. *DNA Res* **22**, 307-318 (2015).
6. Zalzman, M., *et al.* Zscan4 regulates telomere elongation and genomic stability in ES cells. *Nature* **464**, 858-863 (2010).
7. Zhang, Q., *et al.* Tcstv1 and Tcstv3 elongate telomeres of mouse ES cells. *Sci Rep* **6**, 19852 (2016).
8. Dick, S.A., *et al.* Self-renewing resident cardiac macrophages limit adverse remodeling following myocardial infarction. *Nat Immunol* **20**, 29-39 (2019).
9. Molawi, K., *et al.* Progressive replacement of embryo-derived cardiac macrophages with age. *J Exp Med* **211**, 2151-2158 (2014).

Dear Young-sup,

Thank you for submitting a revised version of your manuscript. I have now gone through your response, and I find it generally reasonable. I will therefore be happy to extend official acceptance of your manuscript after implementation of the editorial formatting adjustments listed below:

1. Please make sure that the order of the sections in the manuscript is as follows: abstract, introduction, results, discussion, materials & methods, data availability section, acknowledgments, disclosure statement and competing interests, references, main figure legends, tables, expanded figure legends.
2. Please upload the main figures as individual production quality figure files in the .eps, .tif, or .jpg format (one file per figure).
3. Please remove "Subject terms" and "Abbreviations" from the manuscript text.
4. Please check that the funding information is correct and identical both in the manuscript and our online system. Currently, MSIT No. 2020R1A2C3003784 and No. 2020M3A9I4038454, RS-2024-00333839, RS-2024-00509295 and 2023(6-2023-0168) are missing in our online system. In the manuscript text file, please add funding sources in the "Acknowledgments" section.
5. Please rename "Disclosure" section into "Disclosure and competing interests statement" (further info: <https://www.embopress.org/page/journal/14602075/authorguide#conflictsofinterest>).
6. Please update references according to The EMBO Journal style - where there are more than 10 authors on a paper, the first 10 should be listed, followed by 'et al.' Please see further information here: <https://www.embopress.org/page/journal/14602075/authorguide#referencesformat>
7. Please merge "Code Availability" with Data Availability section. Please also add a resolvable link for GEO dataset. More information about the format of this section can be found here: <https://www.embopress.org/page/journal/14602075/authorguide#dataavailability>.
8. All Materials and Methods need to be described in the main text using our 'Structured Methods' format. According to this format, the Methods section includes a Reagents and Tools Table (listing key reagents, experimental models, software and relevant equipment and including their sources and relevant identifiers) followed by a Methods and Protocols section describing the methods, ideally using a step-by-step protocol format. The aim is to facilitate adoption of the methodologies across labs. Please download and fill our Reagents and Tools Table template (.docx), which you can find in our author guidelines: <https://www.embopress.org/page/journal/14602075/authorguide#structuredmethods>
When submitting your revised manuscript, please do not include the Reagents and Tools Table in the Methods section of the manuscript but upload it as a separate file choosing the file type "Reagent Table".
An example of a Method paper with Structured Methods can be found here: <https://www.embopress.org/doi/10.15252/msb.20178071>.
9. In the Appendix, please remove blue font and upload the file in the .pdf format.
10. Please update the callouts for the appendix tables in the manuscript text to Appendix Table S1-3.
11. During our routine text plagiarism check, we noted that the first sentence in the introduction appears identical to that from another publication - please see the attached screenshot. Please rephrase the text accordingly.
12. Our data editors have flagged the following issues in figure legends that need correcting:
 - Please note provide information on the number and nature of replicates in the legends of figures 3F, 4F.
 - Please define the white arrows in the legend of figures 4A, B; 5D.
13. Papers published in The EMBO Journal are accompanied online by a 'Synopsis' to enhance discoverability of the manuscript. It consists of A) a short (1-2 sentences) summary of the findings and their significance, B) 3-4 bullet points highlighting key results and C) a synopsis image that is 550x300-600 pixels large (width x height, jpeg or png format). You can either show a model or key data in the synopsis image. Please note that the image size is rather small and that text needs to be readable at the final size.

With best wishes,

Ieva

Ieva Gailite, PhD
Senior Scientific Editor
The EMBO Journal
Meyerhofstrasse 1

D-69117 Heidelberg
Tel: +4962218891309
i.gailite@embojournal.org

We realize that it is difficult to revise to a specific deadline. In the interest of protecting the conceptual advance provided by the work, we recommend a revision within 3 months (13th Jul 2025). Please discuss the revision progress ahead of this time with the editor if you require more time to complete the revisions.

The authors addressed the remaining editorial issues.

Dear Young-sup,

Thank you for addressing the final editorial points. I sincerely apologise for the slow process from our side due to the high number of submissions that we experience at the moment. I am now pleased to inform you that your manuscript has been accepted for publication - congratulations on a nice study!

Before we forward your manuscript to our publishers, we would like to propose some edits in the manuscript title, abstract and synopsis (please see below and in the attached text file). I have also written a short blurb that will accompany the title of your manuscript in our online table of contents. Please let me know if any corrections or adjustments are needed:

Title:

CX3CR1+ progenitor cells originate from epiblasts and generate cardiovascular cells during cardiogenesis

Blurb:

Lineage tracing experiments in the mouse identify cardiomyocytes and endothelial cells as the progeny of cardiac macrophage progenitors in the developing heart.

Synopsis:

CX3CR1+ cells form tissue-resident macrophages in the heart that protect against cardiac injury. This study reveals CX3CR1+ cells as a multipotent progenitor population contributing also to cardiomyocytes and heart endothelial cells during early embryonic development.

- CX3CR1+ cells emerge from a subset of epiblasts and subsequently contribute to the parietal endoderm cells.
- Epiblast-derived CX3CR1+ cells contribute to cardiomyocytes and endothelial cells.
- Cardiomyocytes and endothelial cells derived from prenatal CX3CR1+ cells persist in the adult heart.
- Mouse ESC-derived CX3CR1+ cells differentiate into cardiomyocytes and endothelial cells in the adult mouse heart.

If you have any questions, please do not hesitate to contact the Editorial Office. Thank you for this contribution to The EMBO Journal!

With best wishes,

Ieva
